



# Australian wildfire smoke in the stratosphere: the decay phase in 2020/21 and impact on ozone depletion

Kevin Ohneiser[1], Albert Ansmann[1], Bernd Kaifler[2], Alexandra Chudnovsky[3], Boris Barja[4], Daniel A. Knopf[5], Natalie Kaifler[2], Holger Baars[1], Patric Seifert[1], Diego Villanueva[1], Cristofer Jimenez[1], Martin Radenz[1], Ronny Engelmann[1], Igor Veselovskii[6], and Félix Zamorano[4]

[1]Leibniz Institute for Tropospheric Research, Leipzig, Germany
[2]Deutsches Zentrum für Luft- und Raumfahrt, Institut für Physik der Atmosphäre, Oberpfaffenhofen, Germany
[3]Tel Aviv University, Porter School of Earth Sciences and Environment, Tel Aviv, Israel
[4]Atmospheric Research Laboratory, University of Magallanes, Punta Arenas, Chile
[5]School of Marine and Atmospheric Sciences, Stony Brook University, Stony Brook, NY 11794-5000, USA
[6]Prokhorov General Physics Institute of the Russian Academy of Sciences, Moscow, Russia

**Correspondence:** K. Ohneiser (ohneiser@tropos.de)

**Abstract.**

Record-breaking wildfires raged in southeastern Australia in late December 2019 and early January 2020. Rather strong pyrocumulonimbus (pyroCb) convection developed over the fire areas and lifted enormous amounts of biomass-burning smoke into the tropopause region and caused the strongest wildfire-related stratospheric aerosol perturbation ever observed around the globe. We discuss the geometrical, optical, and microphyscial properties of the stratospheric smoke layers and the decay of this major stratospheric perturbation. A multiwavelength polarization Raman lidar at Punta Arenas (53.2°S, 70.9°W), southern Chile, and an elastic-backscatter Raman lidar at Río Grande (53.8°S, 67.7°W) in southern Argentina were operated to monitor the major record-breaking event until the end of 2021. These lidar measurements can be regarded as representative for mid to high latitudes in the Southern Hemisphere. A unique dynamical feature, an anticyclonic, smoke-filled vortex with 1000 km horizontal width and 5 km vertical extent, which ascended by about 500 m per day, was observed over the full last week of January 2020. The key results of the long-term study are as follows: The smoke layers extended, on average, from 9 to 24 km in height. The smoke partly ascended to more than 30 km height as a result of self-lifting processes. Clear signs of a smoke impact on the record-breaking ozone hole over Antarctica in September-November 2020 were found. A slow decay of the stratospheric perturbation detected by means of the 532 nm aerosol optical thickness (AOT) yielded an e-folding decay time of 19-20 months. The maximum smoke AOT was around 1.0 over Punta Arenas in January 2020 and thus two to three orders of magnitude above the stratospheric aerosol background of 0.005. After two months with strongly varying smoke conditions, the 532 nm AOT decreased to 0.03-0.06 from March-December 2020 and to 0.015-0.03 throughout 2021. The particle extinction coefficients were in the range of 10-75 $Mm^{-1}$ in January 2020, and later on mostly between 1 and 5 $Mm^{-1}$. Combined lidar-photometer retrievals revealed typical smoke extinction-to-backscatter ratios of 69±19 sr (at 355 nm), 91±17 sr (at 532 nm), and 120±22 sr (at 1064 nm). An ozone reduction of 20-25% in the 15-22 km height range was observed over Antarctic and New Zealand ozonesonde stations in the smoke-polluted air with particle surface area concentrations of 1-5 $\mu m^2\ cm^{-3}$.



## 1 Introduction

Extremely strong and long-lasting bushfires in southeastern Australia (Boer et al., 2020) in combination with extraordinarily strong pyrocumulonimbus (pyroCb) activity at the end of December 2019 and in the beginning of January 2020 induced a
major perturbation of the stratospheric aerosol conditions (Peterson et al., 2021). The smoke dispersed over most parts of the Southern Hemisphere, including Antarctica (Tencé et al., 2021), and reached heights up to 35 km (Khaykin et al., 2020; Kablick et al., 2020). The Black Summer fire season of 2019–2020 in southeastern Australia, denoted as Australian New Year Super Outbreak (ANYSO) event by Peterson et al. (2021) caused a three times higher stratospheric aerosol mass of injected smoke than the record-breaking Canadian wildfires (Pacific Northwest Event, PNE) in the Northern Hemisphere in
August 2017 (Peterson et al., 2018). The latter led to the largest stratospheric perturbation by smoke over Germany (Ansmann et al., 2018) and Europe (Baars et al., 2019). The ANYSO-related maximum monthly mean aerosol optical thickness (AOT) for the latitudinal belt from 20°S to 70°S exceeded even the maximum AOT (for these southern latitudes) in this latitudinal belt observed after the major eruption of the Pinatubo volcano (Hirsch and Koren, 2021). The stratospheric smoke particles influenced climate conditions (Hirsch and Koren, 2021; Heinold et al., 2021; Stocker et al., 2021) and probably also led to the
record-breaking ozone depletion in Antarctica in September 2020 (Stone et al., 2021).

Wildfire smoke particles, mainly consisting of brown carbon with a few percent of black carbon (Hems et al., 2021), considerably absorb solar radiation and can perturb shortwave and longwave radiative fluxes, and thus dynamical processes. Several Australian smoke plumes were found to significantly alter the dynamic circulation in the lower stratosphere (Khaykin et al., 2020; Kablick et al., 2020; Allen et al., 2020; Lestrelin et al., 2021). The potential of smoke to ascend over several months by
heating the air due to strong light absorption is an important aspect that significantly prolongs the residence time of wildfire smoke in the stratosphere. As Peterson et al. (2021) pointed out, the 'diabatic lofting' effect is often strong enough to oppose the mean downward motion, ultimately increasing plume lifetime in the stratosphere. Several ANYSO smoke ascended from initial injection heights of 14–17 km to about 35 km within 40 days (Khaykin et al., 2020).

The spread of the UTLS (upper tropospheric and lower stratospheric) smoke aerosol and the decay of the stratospheric
perturbation was continuously monitored with the space lidar CALIOP (Cloud Aerosol Lidar with Orthogonal Polarization) of the CALIPSO (Cloud-Aerosol Lidar and Infrared Pathfinder Satellite Observations) mission (Winker et al., 2009) and by means of passive spaceborne remote sensing (Hirsch and Koren, 2021; Kablick et al., 2020; Khaykin et al., 2020; Kloss et al., 2021b). The satellite observations were accompanied by ground-based lidar and AERONET (Aerosol Robotic Network) photometer observations (Ohneiser et al., 2020; González et al., 2020; Ansmann et al., 2021a; Tencé et al., 2021). First results covering the
January 2020 observations were presented in Ohneiser et al. (2020). The multiwavelength polarization Raman lidar techniques as applied at Punta Arenas is used for tropospheric wildfire smoke characterization since more than 20 years (Wandinger et al., 2002; Müller et al., 2005). We expanded the range of applications towards stratospheric smoke after the record-breaking Canadian fires in August 2017 (Haarig et al., 2018), the huge Siberian fires in July-August 2019 (Ohneiser et al., 2021), and the Australian fires (Ohneiser et al., 2020; Ansmann et al., 2021b).





In this article, we present long-term observations conducted with two ground-based Raman lidars at Punta Arenas, Chile, and
Río Grande, Argentina, at the southernmost tip of South America. These measurements can be regarded as representative for the
southern part of the Southern Hemisphere (latitudes $> 40°$S). The paper is organized as follows. Section 2 briefly describes the
field campaigns, lidar stations, instruments, and measurement products. In Sect. 3, the observations are discussed. We start with
the initial phase of the smoke event (Sect. 3.1), provide information on the amount of injected smoke end of December 2019

and in the begining of January 2020, illuminate the initial spread of the stratospheric smoke based on satellite observations,
and highlight a unique observation of a rotating and ascending quasi ellipsoidal smoke field (1000 km in diameter, 5 km thick)
which stayed for more than 10 days close to the southernmost tip of South America (Sect. 3.1.1), and thus close to our lidar
stations. Next, we discuss to measurements that provide an overview of the lidar-derived smoke products in Sect. 3.2. Sect. 3.3
concentrates on the long-term measurements covering two years. The discussion is based on the observed geometrical (layer

base, top, vertical extent), optical properties (backscatter, extinction, depolarization and lidar ratio, aerosol optical thickness,
AOT) as well as microphysical retrievals (particle number, surface area, mass concentrations). The decay behavior of the
stratospheric perturbation is then explicitly studied in Sect 3.5 and compared with other major smoke-related stratospheric
perturbations. In Sect. 4, we discuss the impact of the smoke on the observed record-breaking ozone depletion over Antarctica
in September to November 2020. We continue in this way with the discussion on smoke-induced ozone depletions started in

Ohneiser et al. (2021) and highlighted by Voosen (2021). A summary and concluding remarks in Sect. 5 complete the article.

## 2    Lidar and photometer observations, data analysis, and auxiliary meteorological data

### 2.1    Polly and AERONET observations at Punta Arenas, Chile

Lidar observations were performed at the campus of the University of Magallanes (UMAG) at Punta Arenas (53.2°S, 70.9°W;
9 m above sea level, a.s.l.) in the framework of the DACAPO-PESO (Dynamics, Aerosol, Cloud And Precipitation Observa-

75    tions in the Pristine Environment of the Southern Ocean) campaign lasting from November 2018 to November 2021 (Radenz
et al., 2021). The main goal of DACAPO-PESO was the investigation of aerosol–cloud interaction processes in rather pristine,
unpolluted marine conditions.

    A Polly instrument (*PO*rtab*L*le *L*idar s*Y*stem) (Engelmann et al., 2016) was used for aerosol profiling. This multiwavelength
polarization Raman lidar has 13 channels and continuously measures elastically and Raman backscatter signals at the laser

80    wavelengths of 355, 532, and 1064 nm and respective Raman backscattering wavelengths of 387 and 607 nm (nitrogen Raman
scattering) and 407 nm (water vapor Raman scattering). These observations permit us to determine height profiles of the particle
backscatter coefficient at the laser wavelengths of 355, 532 and 1064 nm wavelength, particle extinction coefficients at 355 and
532 nm, the particle linear depolarization ratio at 355 and 532 nm, and of the water-vapor-to-dry-air mixing ratio by using the
Raman lidar return signals of water vapor and nitrogen. The Polly instrument is designed for automated continuous profiling of

85    aerosols and clouds and thus was running around the clock. Well-defined breaks were necessary to exchange laser flash lamps,
to run different calibration procedures, to check the full setup, and to perform an overall alignment of the Polly instrument.





Details of the stratospheric Polly data analysis including an uncertainty discussion is given in the recent publication of Engelmann et al. (2021) and Ohneiser et al. (2020, 2021). The Raman lidar method was exclusively used to determine particle backscatter and extinction profiles up to the top of the Australian smoke layers. The particle backscatter coefficient is obtained from the ratio of the elastic backscatter signal (355 nm, 532 nm) to the respective Raman signal (387 nm, 607 nm). The 1064 nm backscatter coefficient is calculated from the ratio of the 1064 nm elastic backscatter signal to the 607 nm nitrogen Raman signal. In the retrieval of the extinction coefficient, a least-squares linear regression is applied to the respective Raman signal profiles. To obtain stratospheric background aerosol information at heights above the smoke layer (usually above >25 km height), the backscatter and extinction properties were determined from the elastic backscatter signal profiles by using the so-called Klett method which needs an height-independent extinction-to-backscatter ratio (lidar ratio) as input (Fernald, 1984). We used a lidar ratio of 50 sr in the background aerosol retrieval. Calibration heights were generally set into the clean stratosphere (above 30-35 km height). The backscatter signal intensities were always sufficiently above the detection limits at heights of 30-35 km.

Different expressions for the Ångström exponent were used to characterize the spectral dependence of the aerosol optical properties. The Ångström exponent $A_{x,\lambda_i,\lambda_j} = \ln(x_i/x_j)/\ln(\lambda_j/\lambda_i)$ describes the wavelength dependence of an optical parameter $x$ (backscatter coefficient $\beta$ or extinction coefficient $\sigma$) in the spectral range from wavelength $\lambda_i$ to $\lambda_j$.

The particle linear depolarization ratio (PLDR) was used to characterize the shape properties of the smoke particles. PLDR is obtained from the cross-to-co-polarized signal ratio after correction of Rayleigh contributions to light depolarization. "Co" and "cross" denote the planes of polarization parallel and orthogonal to the plane of linear polarization of the transmitted laser pulses, respectively. The particle depolarization ratio, defined as the ratio of the cross-to-co-polarized backscatter coefficient, was observed at 355 and 532 nm. Irregularly shaped fresh smoke particles cause considerable light depolarization and PLDR values around 0.2, whereas PLDR<0.05 in the case of aged (spherical) smoke particles.

Although the Polly data analysis software package delivers automatically calculated aerosol optical properties, the lidar observations presented in this article were manually analyzed. To keep the uncertainties in the derived aerosol quantities at an acceptably low level of around <10% for the backscatter coefficients and depolarization ratios and of the order of 20–30% in the case of particle extinction coefficients and lidar ratios, large vertical signal smoothing and regression window lengths of 500 to 2500 m and signal averaging times from 30 minutes to several hours had to be applied. More details are given in Ohneiser et al. (2021).

To obtain estimates of microphysical properties of the smoke particles such as mass, volume, and surface area concentration, a conversion method is available (Ansmann et al., 2021a). The 532 nm particle backscatter coefficients are input in the retrieval process. In the case of the Australian smoke observations, the backscatter coefficients are converted to extinction profiles, in the first step, by assuming a smoke lidar ratio of 85 sr at 532 nm. In the second step, the extinction coefficients were converted to microphysical properties by respective smoke conversion factors. To obtain smoke mass concentrations, we assume a smoke particle density of 1.15 g cm$^{-3}$ (Ansmann et al., 2021a).

Auxiliary data were required in the lidar data analysis in form of temperature and pressure profiles in order to calculate and correct for Rayleigh backscatter and extinction influences on the measured lidar return signal profiles. For auxiliary meteorolog-





ical observations, we used GDAS1 (Global Data Assimilation System 1) temperature and pressure profiles with 1° horizontal resolution from the National Weather Service's National Centers for Environmental Prediction (GDAS, 2021). We checked the quality and accuracy of the GDAS1 profiles through a comparison with respective temperature and pressure profiles measured

with radiosonde at Punta Arenas airport (daily launch at 12:00 UTC). CALIOP, Sentinel-5, and MODIS (Moderate Resolution Imaging Spectroradiometer) data (CALIPSO, 2021b; ERA5, 2021; MODIS, 2021a, b, c) were used to support the complex study of the long-range smoke transport across the Pacific Ocean.

Besides the continuous lidar measurements, an AERONET sunphotometer was operated (Holben et al., 1998; AERONET, 2021). Measurement wavelengths and retrieval products of the Punta Arenas sunphotometer is given in Ansmann et al. (2021a).

Sunphotometer aerosol optical depths measured at 380, 500 and 1020 nm were partly combined with column-integrated backscatter coefficients from lidar to estimate stratospheric lidar ratios at 355, 532 and 1064 nm. This approach is described in detail in Sect. 3.3.

To study the impact of the stratospheric wildfire smoke on the stratospheric ozone layer, we made use of the long-term Southern Hemispheric ozone data record (NDACC, 2021). Ozonesonde observations are regualrly performed by the Network

for the Detection of Atmospheric Composition Change (NDACC).

## 2.2 CORAL observations at Río Grande, Argentina

As part of the SouthTRAC-GW (Southern Hemisphere Transport, Dynamics, and Chemistry-Gravity Waves) mission (Rapp et al., 2021), the Compact Rayleigh Autonomous Lidar (CORAL) (Kaifler and Kaifler, 2021) was operated at Estación Astrónomicas Río Grande (53.8°S, 67.7°W; 9 m above sea level, a.s.l.), which is close to the airport of Río Grande, in southern

Argentina. The station is 230 km east to southeast of Punta Arenas. The lidar is optimized for density and temperature measurements in the middle atmosphere, approximately between 15 km and 90 km. During the stratospheric smoke period, the powerful lidar delivered valuable smoke optical properties at 532 nm for heights from 15 to 30 km. CORAL operates autonomously during clear sky conditions in darkness and uses three height-cascaded elastic-backscatter channels (532 nm wavelength) and one Raman channel (607 nm wavelength) which allows for the independent retrieval of the 532 nm backscatter coefficient, extinc-

tion coefficient, and lidar ratio. Meteorological data is used from radiosonde launches at Punta Arenas airport (daily launch at 12:00 UTC) (Uni-Wyoming, 2021).

We applied the Polly data analysis software to analyze the CORAL raw signal counts (at 532 and 607 nm) to avoid any bias caused by different data analysis concepts. We noticed and corrected a small cross-talk effect in the Raman channel (less than 0.8%). The signal reference height was set to $30\pm1$ km with a reference particle backscatter coefficient of $0.001 \text{Mm}^{-1}$.

Vertical signal smoothing and regression windows (extinction and lidar ratio retrieval) was set to 2100 m. Profiles of the particle backscatter coefficient were obtained with smoothing lengths of 500 m.



## 3   Results

### 3.1   The major stratospheric perturbation in January 2020

According to Peterson et al. (2021), the Black Summer fire season of 2019–2020 in southeastern Australia contributed to
a rather intense period of fire-induced and smoke-infused thunderstorms. ANYSO resulted in roughly 1.0 Tg of cumulative
smoke particle mass being injected into the lower stratosphere. It was characterized as a pyroCb 'super outbreak' because of
its exceptional scale and magnitude. An unusually large number of 38 distinct convective cloud towers, denoted as pyroCb
pulses, developed over a prolonged period of 51 non-consecutive hours. ANYSO occurred in two distinct phases, with the first
and largest occurring during 29–31 December with an overall duration of about 45 h (from 09:30 UTC on 29 December to
6:40 UTC on 31 December 2019). During the first phase 33 distinct pyroCb pulses injected smoke into the UTLS. Part of the
smoke reaching the upper part of the troposphere was further lifted to UTLS heights by strong convective activity in the outflow
regime over the Pacific Ocean on the way to New Zealand (Hirsch and Koren, 2021) and by large cyclonic systems that formed
further downstream east of New Zealand and facilitated a direct transport of smoke into the lower stratosphere (Magaritz-Ronen
and Raveh-Rubin, 2021). The second phase of ANYSO began on 4 January 2020, after 3 days without pyroCb activity. The
duration of this pyroCb activity (with 5 pyroCb pulses) was more typical of previous significant events (such as the Canadian
wildfires, PNE, British Columbia, 12-13 August 2017) (Peterson et al., 2018), spanning just over six hours. PNE comprised
less than 10 pyroCb pulses, however caused the largest smoke-related stratospheric perturbation ever observed over Europe
(Ansmann et al., 2018; Baars et al., 2019).

ANYSO's first phase stands as the largest known stratospheric injection of smoke particles linked to a distinct period of
pyroCb activity (0.2–0.8 Tg) (Peterson et al., 2021). ANYSO's second phase injected an estimated 0.1–0.3 Tg of additional
smoke particle mass into the lower stratosphere. The cumulative smoke particle mass injected into the stratosphere by both
phases of pyroCb activity was thus 0.3–1.1 Tg and therefore at least three times larger than the PNE smoke mass of 0.3 Tg
(Peterson et al., 2018). More than half of the 38 observed pyroCbs pulses injected smoke particles directly into the stratosphere.
Taking maximum updraft velocities of 35-60 m s$^{-1}$ in these cumulus towers into account (Rodriguez et al., 2020), smoke can
be lifted within less than an hour from the surface to the lower stratosphere. The UTLS smoke plumes encircled a large swath
of the Southern Hemisphere while continuously rising due to local heating by smoke absorption of solar radiation. More details
to the asent of the smoke is provided in the following subsection.

Figure 1 provides an overview of the smoke situation over the southern Pacific and Atlantic Oceans in January 2020 (MODIS,
2021a, b, c). Figure 1a shows an optically dense smoke field between Australia (Melbourne) and New Zealand (Wellington) on
5 January 2020. In Fig. 1b, monthly mean January AOT (500nm) values are visualized, frequently clearly exceeding 0.5, and
sometimes 1.0 between Australia and South America. The smoke was mainly transported from Australia across northern New
Zealand down to southern South America and Antarctica during the initial phase of the long lasting stratospheric perturbation.
Figure 1c and d zoom into the smoke situation close to the source region, where the 10-d AOT mean values were of the order
of 1.0 to >2.0 during the injection phase (1-10 January 2020), and after long range transport over 10000 km southwest of
southern South America, were still high 10-d mean AOT values of the order of 0.4-0.8 were found.



Figure 1e provides time series of daily AOT observed with MODIS (Terra and Aqua satellites), AERONET photometer and lidar at Punta Arenas. Initially, the area mean AOT values were close to 2.0 in the black rectangle shown in Fig. 1c and then mostly in the range from 0.2 to 0.5 after 10 January 2020. According to the time series of daily, cloud-screened MODIS AOT observations within the black rectangle in Fig. 1d, the rotating smoke-filled vortex close to Punta Arenas caused AOTs of

0.3–0.5 during the last week of January 2020. Details to the rotating smoke fields are given in the next subsection. Note that the Southern Ocean AOT of the clean marine troposphere is of the order of 0.03-0.05 at 500 nm, as indicated by the Punta Arenas AERONET sunphotometer observations from 1-6 January 2020.

Nighttime Raman lidar observations of 532 nm extinction profiles are used in Fig. 1e to compute the lidar-derived stratospheric AOTs. First smoke plumes reached Punta Arenas on 5-7 January 2020. Significantly enhanced stratospheric AOTs were

then observed on 8 January 2020 (Ohneiser et al., 2020). AOT values observed with lidar (during nighttime) and photometer (during daytime) slowly increased towards values around 0.3-0.35 in the last week of January 2020. A few values measured with CALIOP during smoke vortex overflights (close to South America) from 23-31 January 2020 are included in Fig. 1e.

### 3.1.1 The unique self-organized, rotating and ascending smoke-filled vortex

A new atmospheric phenomenon was detected in January-March 2020 (Kablick et al., 2020; Khaykin et al., 2020; Allen et al.,

2020; Lestrelin et al., 2021). It coupled smoke occurrence, aerosol-radiation interaction, and dynamical processes. The striking discovery was the observation of several stratospheric smoke plumes that self-organized as quasi-ellipsoidal anticyclonic vortices, persisted over weeks to months, and traveled around the Southern Hemisphere. These plumes transported smoke up to heights above 30 km. The most intense, self-maintained vortex measured about 1000 km in diameter and about 5 km in vertical extent. This highly stable vortex persisted in the stratosphere for over 13 weeks, crossed the Pacific within two weeks

and hovered above the tip of South America for more than a week. It then followed a 10 week westbound round-the-world journey that could be tracked over 66000 km until the beginning of April 2020 (Khaykin et al., 2020). The vortex lifted a confined bubble of smoke and moisture to 35 km altitude (Khaykin et al., 2020), an altitude not reached by tropospheric aerosols since the Pinatubo eruption (McCormick et al., 1995; Stenchikov et al., 2021). Aerosol heating was essential in maintaining the structure and providing the lift. In turn, the vortex created a confinement that preserved the embedded smoke cloud from

being rapidly diluted within the environment (Lestrelin et al., 2021). Allen et al. (2020) emphasized that the peculiar motion was related to the steady rise in plume potential temperature of about 8 K per day in January and 6 K per day in February, due to local heating by smoke absorption of solar radiation. This heating resulted in an anticyclonic (i.e., counterclockwise) circulation with winds rotating around the plume at about 15 m s$^{-1}$ (Kablick et al., 2020). Kablick et al. (2020) highlighted the importance of this detection: This is the first evidence of smoke causing changes to winds in the stratosphere and opens up

a whole new vein of scientific research.

The rotating smoke field stayed close to the southernmost tip of South America (see also Fig. 1d) for 10-12 days, until the beginning of February 2020, and ascended by about 500 m per day. During the steady and monotonic ascent, the smoke reached heights with dominating easterly winds. The center of the vortex was closest to Punta Arenas on 29-30 January 2020.





Figure 2a shows the disk-like structure southwest of the two lidar stations at Punta Arenas and Río Grande. The Sentinel-
5 UV aerosol index from 340 and 388 nm is shown for 26 January 2020 (Sentinel-5, 2021). Bright blue colors indicate the
presence of absorbing aerosol. The smoke plume was well captured by CALIPSO lidar observation on 25 January 2020, as
shown in Fig. 2b (dashed curve in (a) shows the CALIOP track). The smoke vortex was detected between 19 and 26 km height
at latitudes from 52°S to 65°S.

According to the Punta Arenas lidar observations in Fig. 3, parts of the rotating smoke field covered southern Chile from
24-31 January 2020. The flat base of the dome-like plume structure (see Fig. 2b) ascended from 19 km on 24 January to
22 km on 30 January and thus with a speed of about 500 m per day over the Polly lidar site. This is in agreement with the
CALIOP observations. Khaykin et al. (2020) found that the initial ascent rate was at 0.45 km day$^{-1}$, with an average rate of
0.2 km day$^{-1}$ for the 3-month period during which this smoke field existed.

We performed a detailed analysis of the Punta Arenas lidar observations on 29 January 2020 when the vortex center was
closest to the Polly lidar site. Table 1 summarizes the essential optical and microphysical properties discussed in detail by
Ansmann et al. (2021a). The very large 532 nm lidar ratio of 106 sr points to strongly light-absorbing smoke particles. The
mass, volume, surface area, and number concentrations indicate a strong perturbation of the usually rather clean stratosphere.
At unperturbed conditions, particle extinction coefficients are of the order of 0.1 Mm$^{-1}$ (Sakai et al., 2016) and particle surface
area concentrations are in the range of 0.5-1 $\mu$m$^2$ cm$^{-3}$ (Hofmann and Solomon, 1989; Deshler et al., 2003). Volume size
distributions for the Australian smoke, obtained from lidar data inversions (Veselovskii et al., 2002), presented in Ansmann
et al. (2021b); Ohneiser et al. (2021) showed a pronounced accumulation mode (radius range from 0.15-0.75 $\mu$m) with an
effective radius of 0.28-0.29 $\mu$m. The revealed refractive index values and the low single scattering albedo of close to 0.8 at all
three wavelengths (not listed in the table) again point to strongly light absorbing particles.

As discussed by Ohneiser et al. (2020), the lidar ratios for Australian smoke at 532 nm were considerably higher than the
ones for Canadian smoke observed in August 2017 of 60-80 sr (Haarig et al., 2018). The difference is probably related to the
different burning material. Australian's forests mainly consist of eucalyptus trees, whereas western Canadian tree types are
predominantly pine, fir, aspen, and cedar.

## 3.2 Case studies from January 2020: Optical fingerprints of aged wildfire smoke

To provide an overview of the basic lidar products, we begin with two case studies presented in Figs. 4 and 5. The Polly
lidar monitored the smoke layers in terms of particle backscatter coefficients at three wavelengths and the particle extinction
coefficient, lidar ratio, and depolarization ratio at two wavelengths. This set of optical properties allow us to characterize
the smoke in large detail. However, the Polly instrument is optimized for tropospheric aerosol monitoring up to 10-12 km
height so that extinction and lidar-ratio profiling was already at its limit for heights above 14 km, especially in the case of the
measurement wavelengths of 355 and 387 nm for which signal attenuation by Rayleigh scattering is strong and signal quality
therefore gets quickly low with height. On the other hand, the powerful CORAL instrument was optimized for stratospheric
and lower mesospheric observations up to 90 km height and thus shows good performance at greater heights as demonstrated





in Fig. 5. Unfortunately, the minimum measurement height of CORAL is about 15 km which makes direct comparisons with the Polly observations problematic.

The measurements shown in Fig. 4 were performed when the first strong smoke plumes reached southern South America
(Ohneiser et al., 2020). The maximum backscatter and extinction values were found around 13-14.5 km height, layer base and top were at 12.7 and 14.7 km. The maximum extinction coefficients in the center of the smoke layer were around 70 Mm$^{-1}$ at 532 nm and 120 Mm$^{-1}$ at 355 nm. The 532 nm AOT was 0.08. The enhanced particle depolarization ratios of about 0.2 (355 nm) and 0.15 (532 nm) are clear indications for wildfire smoke reaching the stratosphere via fast pyroCb lifting (Haarig et al., 2018; Ohneiser et al., 2020).

In Fig. 5, the Polly and CORAL measurements are compared. The lidar observations on 26 January 2020 were performed close to the edge of the rotating and ascending smoke-filled vortex. The two lidars (230 km apart from each other) saw different parts of this horizontally inhomogeneous smoke field. Nevertheless, good agreement was found for the smoke extinction-to-backscatter ratio at 532 nm. The more powerful CORAL instrument was able to provide high-quality extinction profiles even in the upper part of the smoke layer, extending from 19.5-23 km height according to the backscatter profiles in Fig. 5a.
Stratospheric AOTs at 532 nm were 0.16 (Punta Arenas) and close to 0.05 (Río Grande).

Based on Figs. 4 and 5 we can summarize the following optical fingerprints of aged wildfire smoke. This aerosol type shows a clear wavelength dependence of the particle backscatter coefficient and a less pronounced spectral dependence of the extinction coefficient. Consequently, the extinction-to-backscatter ratio shows an inverse spectral behavior. The 532 nm lidar ratios are considerably larger than the 355 nm lidar ratios. Furthermore, the 532 nm lidar ratios typically exceed 70 sr and can
be even larger than 100 sr. The CORAL observations yielded lidar ratios in the range from 75-85 sr on five different days in January and February 2020, but also a case with 100 sr on 16 January 2020. As mentioned already, the depolarization ratio of typically 0.15-0.2 at 532 nm and 0.2-0.25 at 355 nm are clear indications for smoke lifted into the stratosphere via strong pyroCb convection, i.e., within a rather short time period so that the morphological properties of the injected irregularly shaped carbonaceous particles remained widely unchanged when the particles entered the dry stratosphere. Only a small fraction of the
smoke serves as cloud condensation nuclei and/or ice-nucleating particles, and the lack of significant precipitation produced in these cloud towers implies that only a small fraction of the smoke is scavenged, so that most of it is exhausted through the anvil to the upper troposphere and lower stratosphere (Rosenfeld et al., 2007).

### 3.3 The decay phase: observation over two years (2020-2021)

Figure 6 provides an overview of the stratospheric perturbation from January 2020 to November 2021 as observed with the
two ground-based lidars. One set of lidar products per day is considered. Gaps in the data time series are caused by fog, and low-cloud events, and by instrumental problems (long gap in July 2021). In the case of the CORAL observations, the laser stopped working properly in January 2021. Fortunately, these observations cover the main phase of the major stratospheric perturbation. Because of the designed lidar configuration, CORAL provided data for heights >15 km only.

In Fig. 6a and b, layer height, depth and particle light-extinction information is given. We determined the layer base and
top heights by visual inspection. In the case of Polly, the mean height profile of the 1064 nm backscatter coefficient was used



which is rather sensitive to particle backscattering. The base height is defined as the altitude at which the 1064 nm backscatter coefficient started to increase in the middle to upper troposphere after a minimum in the lower to middle free troposphere. The layer top was set to the height where the total-to-Rayleigh backscatter ratio at 1064 nm dropped below a value of 1.1. A similar approach was applied to the elastic-backscatter signals, measured with CORAL at 532 nm, to determine the smoke

layer top. The vertical bars in Figs. 6a and b are colored to distinguish different levels of the aerosol loading expressed in terms of the particle extinction coefficient. The backscatter coefficients at 532 nm were multiplied by a lidar ratio of 85 sr to obtain the extinction coefficients. A lidar ratio of 85 sr represents well the mean extinction-to-backscatter relationship of the entire Australian smoke data set. The tropopause is given in Fig. 6a to indicate that smoke was also present in the upper troposphere and influenced cirrus formation (Knopf et al., 2018; Ansmann et al., 2021a; Engelmann et al., 2021). The smoke

layers extended from 8-10 km height up to 21-24 km height on most of the days. On average, the layer base and top heights were at 9.8 km and 22.5 km, respectively. No long-term trend in the layer geometrical properties is visible. The optically densest part was found between 12 and 18 km height. The light-extinction values slowly decreased with time (from red to orange to blue Fig. 6a and b). The periods with isolated, probably rotating and ascending smoke-filled vortices are highlighted by big oval circles. Isolated smoke features vanished in the beginning of May 2020 (four months after the ANYSO smoke injection

period).

Figure 6c presents the time series of 532 nm AOT measured at Punta Arenas over the two year period. The AOT values were obtained by integrating the extinction values, i.e., the backscatter values multiplied wiht 85 sr, from layer base to top. The directly determined extinction coefficients (from the Raman signal profiles) were too noisy and in many cases even not available for the upper part the smoke layers, and thus did not permit a trustworthy computation of the AOT time series for

the two-year period. The 355 nm AOT values in Fig. 6c were obtained from the 355 nm backscatter coefficients multiplied with a lidar ratio of 55 sr. Most of the variability in the 355 nm AOTs is caused by signal noise. The almost noise-free 532 nm AOT from the CORAL observations above 15 km is given in Fig. 6c as well. A strong increase of the 532 nm AOT from November-December 2019 to January 2020 and large variability in AOT were found in January 2020 with the two lidars. The AOT reached values up to 1.0 at 532 nm in mid-January over Punta Arenas. The day-by-day AOT variability significantly

decreased in February 2020. Since then, a slow and coherent decrease of the stratospheric perturbation is visible. The AOT value at 532 nm decreased to 0.03-0.06 (February-June 2020), 0.02-0-05 (July-December 2020), and were about 0.015-0.03 in 2021. The AOT for undisturbed, clean stratospheric background conditions for the height range from the tropopause to 30 km height is about 0.005.

The contribution of Australian smoke to the total AOT is shown in Fig. 6d. In order to determine the smoke fraction caused

by the ANYSO smoke injections, we need to consider the natural stratospheric background conditions as well as further contributions by aerosols injected in 2019 (Kloss et al., 2021a, b). In Fig. 6c, different stratospheric background levels for the wavelength of 532 nm are shown as dotted and dashed lines. In November-December 2019, UTLS AOTs around 0.01 (8-30 km height) and stratospheric AOTs around 0.004 (15-30 km height) were observed with the lidars at Punta Arenas and Río Grande, respectively, in reasonable agreement with results from passive remote sensing for heights >15 km (Kloss et al.,

2021b). According to Sakai et al. (2016), who analyzed Japanese lidar observations at Lauder, New Zealand, from 1992 to





2015, the minimum stratospheric background level is reflected in a 532 nm AOT of 0.0025 for the height range from 15-30 km height. The AOT is obtained from the measured minimum column-integrated backscatter coefficient multiplied by a lidar ratio for sulfate aerosol of 50 sr. This minimum background level (for the 15-30 km height range) is indicated by a dark green line in Fig. 6c. From the New Zealand lidar observations, it can be concluded that the UTLS AOT is roughly 0.005 during
clean conditions (light green dotted line in Fig. 6c). This estimate was found to be in agreement with the analysis of satellite observations from 1990 to 2010 during volcanic quiescent times (Solomon et al., 2011).

However, further contribution to the AOT measured in November-December 2019 need to be added. The Ulawun volcano in Papua New Guinea (5°S, 151°E) erupted on 26 June 2019 (injection of 0.14 Tg $SO_2$ into the stratosphere) and on 3-4 August 2019 (injection of 0.3 Tg $SO_2$) (Kloss et al., 2021a) and caused most probably a stratospheric AOT at 532 nm of around 0.005
at the end of September 2019 and of 0.0025 two to three months later. We estimated the Ulawun-related AOT by comparing $SO_2$ emissions and resulting stratospheric sulfate AOT for the Raikoke, Sarychev, and Ulawun eruptions (Haywood et al., 2010; Kloss et al., 2021a, b; Ohneiser et al., 2021). For the height range >15 km, we arbitrarily assumed a volcanic AOT contribution of 0.001 for the November-December 2019 period. To realistically consider the decrease of the Ulawun-related disturbance of the stratospheric aerosol conditions we assumed a decay time of 180 days.

As a second additional contribution Kloss et al. (2021b) identified Australian smoke emitted from September to December 2019. To match the measured AOT values of 0.005 and 0.01 over Río Grande and Punta Arenas, respectively, this contribution must have been also on the order of 0.0025 (for the UTLS height range) and 0.001 (for the height range >15 km). Here, we assume a decay time of one year (365 days). The sum of all three contributions (constant minimum background level plus volcanic AOT plus Australian smoke for the September to December period) is shown as dashed lines in Fig. 6c. By considering
these enhanced time-dependent background levels, we can conclude that the ANYSO smoke caused roughly a factor of 4 higher stratospheric AOT values over the mid latitudes in the Southern Hemisphere for more than one year. Accordingly, the ANYSO smoke fraction for the UTLS height range, shown in Fig. 6d, was about 80% during 2020 and 70%-80% in 2021. Contrary, the Ulawun-related AOT fraction was about 25% in November-December 2019, negligible in January 2020, about 5% in February-April 2020, and <3% since June 2020.

In Fig. 7, the stratospheric aerosol perturbation in 2020 and 2021 is presented as a function of height. Besides the January 2020 mean extinction profile (in red), the February-June 2020 mean profile (orange), and further three half-year mean extinction profiles (blue, green, black) are shown together with the pre-ANYSO 532 nm extinction profile (November-December 2019 in dark grey). The daily height profiles of the particle backscatter coefficient at 532 nm were multiplied by a representative smoke lidar ratio of 85 sr and then separately averaged for the different time periods. In the case of the November and December
2019 backscatter height profiles (pre-ANYSO profiles) we applied a lidar ratio of 60 sr to consider the contribution of volcanic aerosol (lidar ratio of 45 sr) and stratospheric background aerosol (lidar ratio of 50 sr) in addition to the September-December 2019 smoke from Australia (lidar ratio of 85 sr). The dashed lines in Fig. 7a above approximately 25 km height are calculated by applying the Klett method to the strong elastic-backscatter signals (Fernald, 1984) as described in Sect. 2.1. The stratospheric minimum background extinction levels (Sakai et al., 2016) are given as well in Fig. 7 as thin solid-dotted grey line.



The extinction profile in November-December 2019 indicates already significantly enhanced aerosol pollution levels caused by Australian fires since September 2019 (Kloss2021b) and volcanic sulfate aerosol (Ulawun eruption) (Kloss et al., 2021a). Then, in January 2020, the aerosol extinction coefficient increased by 1-2 orders of magnitude compared to the November-December 2019 extinction levels in the height range from 8-25 km, and more than two orders of magnitude with respect to the minimum perturbation (Sakai et al., 2016). The structure of the January 2020 profile is dominated by the arrival of several

single, very dense smoke layers such as the rotating and ascending smoke-filled vortex in the last January week. Monthly mean extinction values of 10-30 $\text{Mm}^{-1}$ were found in January 2020. The following half-year mean extinction profiles showed much lower values. However, these profiles indicate a quite slow decay of the perturbation from February 2020 to November 2021. In the second half of 2021, the extinction values were still a factor of 5-10 higher then the extinction levels representing a minimum stratospheric aerosol load (Sakai et al., 2016). Note also, that the extinction levels were also clearly enhanced

between 26 and 30 km height in 2020-2021.

### 3.4    Smoke intensive parameters

Figure 8 shows time series of smoke intensive properties for the year 2020. The backscatter-related Ångström exponents (defined in Sect. 2.1), the extinction-to-backscatter ratio, and the particle linear depolarization ratio enable a good optical characterization of aged wildfire smoke. The strong variability in the data, especially in the case of the backscatter-related Ångström

exponent for the short-wavelength range (355-532 nm), is caused by signal noise rather than changes in the microphysical and chemical properties of the smoke particles. Mean values of the Ångström exponents and lidar ratios shown in Figure 8 are given in Table 2.

As can be seen in Fig. 8a, the Ångström exponents for the 355-1064 nm and the 532-1064 nm wavelength ranges accumulated between 1 and 2. Such values were already observed in the stratosphere for aged Canadian smoke observed over central Europe

in August 2017 (Haarig et al., 2018) and for aged Siberian smoke in the High Arctic (Ohneiser et al., 2021), and points to a pronounced accumulation-mode-dominated particle size distribution and the absence of a particle coarse mode. The less noisy Ångström exponent for the long wavelength range (532-1064 nm) increases slightly with time which reflects a shift of the size distribution towards smaller particles. Larger particles may have been removed by sedimentation processes.

Lidar ratios at 355 and 532 nm could be measured directly with the two lidars in January and February 2020 only (see

Fig. 8b). The lidar ratios measured with Polly in January 2020 (blue and light green) are also given in Table 1 in Ohneiser et al. (2020). The values range from 50-100 sr (355 nm) and 75-115 sr (532 nm). The CORAL values for 532 nm in Fig. 8b accumulate around 80 sr. Since March 2020, the Raman signal profiles were too noisy and the smoke extinction coefficients too low and did no longer allow a lidar-ratio retrieval. In order to obtain some snapshot-like estimates for the smoke lidar ratio at 355 and 532 nm and even for the IR wavelength of 1064 nm, we combined stratospheric AOT estimates from AERONET

sunphotometer observations at 380, 500, and 1020 nm with respective lidar-derived stratospheric column-integrated backscatter values at 355, 532, and 1064 nm. In this approach, we estimated the stratospheric AOT from the total (tropospheric + stratospheric) AOT measured the AERONET photometer at Punta Arenas during favorable, stable, and temporally constant aerosol conditions. We subtracted the tropospheric AOT contribution from these AERONET AOT values. The tropospheric





AOT was estimated by using the tropospheric backscatter and extinction profiles (from the surface to the tropopause) obtained
with the Fernald method applied to the elastic backscatter signal profiles (Fernald, 1984). We assumed reasonable tropospheric
lidar ratios of 30, 40, and 50 sr for a mixture of continental and marine aerosols in the Fernald data analysis to obtain the
tropospheric AOTs at 355, 532, and 1064 nm. We combined the lidar observations with the near-range and far-range telescope
(receiver units) to obtain the full extinction profile from close to the ground up to the tropopause. In order to transfer the
total (tropospheric + stratospheric) AERONET AOT values at 380, 500, and 1020 nm to the ones for the lidar wavelengths
at 355, 532, and 1064 nm, we used climatologically mean tropospheric Ångström values measured over Punta Arenas during
undisturbed stratospheric aerosol conditions from January to July 2019. The ratio of the stratospheric AOT (from the combined
AERONET and Polly observations) to the respective stratospheric column backscatter coefficient is shown in Fig. 8b. The label
532-A40 indicates, e.g., the use of AERONET observations (index A) and the use of a lidar ratio of 40 sr (index 40) in the
retrieval of the tropospheric AOT at 532 nm. In Fig. 8b, we only show smoke lidar ratio solutions for 355 and 1064 nm for the
tropospheric lidar ratio assumption of 40 sr to avoid overloading of the figure with two many results.

One should emphasize that these retrieved stratospheric lidar ratio must be regarded as rough estimates. In this combined
AERONET-Polly approach we ignore an underestimation of the stratospheric smoke AOT from AERONET sunphotometer
observations up to 20-30% as discussed by Ansmann et al. (2018). Furthermore, measurements with lidar and photometer
did not exactly cover the same time periods and thus may have been conducted at slightly different boundary-layer, free-
tropospheric and stratospheric aerosol conditions. Thus, the shown lidar ratios provide only a very general view on the spectral
behavior of the smoke lidar ratios. Nevertheless, the findings are in line with previous observations (Haarig et al., 2018). When
averaging all lidar ratios obtained with the different methods the resulting lidar ratios are 69 sr, 91 sr, and 120 sr for 355 nm,
532 nm, and 1064 nm, respectively.

Table 2 summarizes the observations in Fig. 8a and b. With the mean lidar ratios 69 sr and 91 sr, the lidar-ratio-related
Ångström exponent $A_{\mathrm{LR}}$ was calculated and also the extinction-related Ångström exponent $A_\sigma$ with $A_\sigma = A_{\mathrm{LR}} + A_\beta = 0.64$. The
low values of $A_\sigma$ and the negative value of $A_{\mathrm{LR}}$ are clear optical fingerprints of wildfire smoke (Ohneiser et al., 2020; Haarig
et al., 2018).

Figure 8c shows the evolution of the particle linear depolarization ratio (PLDR) at 355 nm and 532 nm. High quality depo-
larization ratio observations were frequently only possible in the lower half of the smoke layer. The shown values may thus
not be at all representative for the entire layer (from base to top). Initially, the PLDR showed values close to 20% for fresh
smoke in the stratosphere. In the first two months, the smoke PLDR was very variable with values from almost 0% to 20%.
After 2 months the PLDR decreases to values below 5%. After 3 months, only a few useful measurements were possible and
the PLDR values were below 5% for both wavelengths. The smoke particles obviously became rather compact and spherical
in shape within a short time of about 4-6 weeks. The depolarization ratios are further discussed in the next section.

**3.5 Comparison of three major stratospheric smoke events: ANYSO (2020) vs PNE (2017) vs SILBE (2019)**

In this section, the decay behavior of the stratospheric perturbation caused by the Australian fire smoke is discussed in more
detail. We use the opportunity to compare the ANYSO event with respective observations of two major smoke events occurring





in the Northern Hemisphere in 2017 and 2019-2020 (see Fig. 9). The record-breaking Canadian fire storm (PNE, 2017) was already mentioned in Sect. 3.1. Strong smoke injection into the UTLS regime occurred during a cluster of about 5 pyroCb
pulses over British Columbia, Canada, on 12-13 August 2017 (Peterson et al., 2018, 2021). Lidar network observations of aged smoke after long-range transport over weeks to months were conducted over Europe (indicated by a black circle in Fig. 9) from August 2017 to January 2018 (Baars et al., 2019). We further include lidar observations of the decay phase of a dense stratospheric smoke layer observed over the High Arctic with a Polly lidar aboard the ice breaker Polarstern. These observations from October 2019 to May 2020 were performed in the framework of the MOSAiC (Multidisciplinary drifting Observatory for
the Study of Arctic Climate) expedition (Engelmann et al., 2021; Ohneiser et al., 2021). Extremely strong fires occurred over central and eastern Siberia, north and northeast of Lake Baikal (SIberian Lake Baikal Event, SILBE) from mid July to mid August 2019. The fire smoke reached the UTLS height range most probably by so-called self-lifting processes within 2-7 days after injection into the lower troposphere (Ohneiser et al., 2021). PyroCb activity was absent over the huge Siberian fire region in July and August 2019. Since the Siberian smoke was injected into the UTLS height range about 2500-3500 km south of the
High Arctic, the plumes had to travel a short distance only with the prevailing westerly to southerly winds.

In Fig. 10, the three major stratospheric smoke events are compared regarding their geometrical and optical properties. As can be seen, the vertical extent of the Canadian smoke layers was of the order of 1-4 km and these smoke layers were clearly observable over Europe with lidar up to 150 days after injection only. The Siberian smoke formed an 7-10 km thick layer and was monitored from day 80 up to about 300 days after injection. From December 2019 to May 2020 the smoke was trapped
around the North Pole under the influence of a rather strong polar vortex. The stratospheric perturbation could be observed until the strong polar vortex collapsed in May 2020 so that the smoke became dispersed over a larger region towards the mid latitudes in late spring and summer of 2020. The Australian fires were by far the strongest and produced a 10-15 km thick layer. These smoke layers were clearly detectable even two years after injection. Unfortunately, we had to stop the DACAPO-PESO campaign and thus our Punta Arenas smoke observation end of November 2021. The CORAL system, on the other hand, is
measuring again since December 2021.

The layer base height of the ANYSO and SILBE smoke layers was always close to or slightly below the tropopause (mostly at 8-10 km height). In contrast, the base height of the PNE smoke layer was typically several kilometers above the tropopause. The top height was mostly around 18 km (PNE), 16 km (SILBE) and 23 km (ANYSO). The different top heights had a sensitive impact on the decay of the stratospheric perturbation described in terms of e-folding decay times.

In Fig. 10b, the AOT data sets of the three wildfire events are compared and the decay behavior of the different stratospheric perturbations are presented. As expected, the PNE AOTs were, on average, three to four times lower than the ANYSO AOTs (see discussion in Sect. 3.1). In order to see a clear reduction of the AOT with time the shown PNE and ANYSO AOTs were computed from the smoke extinction coefficients between 13 and 30 km height, and thus for a height range several kilometers above the tropopause. The selection of the 13 km level as base height was necessary in the case of the ANYSO AOTs to
avoid the impact of new aerosol plumes in 2020 and 2021 that regularly appeared in the tropopause region, on the decay time computations. The decay behavior is indicated by straight lines in Fig. 10b. The respective e-folding decay times are given as numbers in Fig. 10b and were computed by using the AOT time series from day 40 to 160 (PNE) and from day 60 to 560



(ANYSO). The first 40-60 days (after smoke injection) were excluded because they showed a rather strong AOT variability and no clear trend.

The MOSAiC observations started about 80 days after injection of the smoke into the UTLS regime so that a removal of strongly varying AOT values in the decay time calculations was not necessary in the case of the Siberian smoke. Furthermore, the entire smoke layer from base to top was considered in the AOT calculations. The impact of tropospheric aerosol sources on the smoke AOT evolution over the High Arctic was found to be negligible. In terms of the total AOT (from layer base to top) the ANYSO and SILBE smoke events were similar regarding the injected smoke amount and corresponding stratospheric

AOTs.

Strong differences in the decay behavior of the stratospheric perturbation in the Northern and Southern Hemisphere were found. A very large value of the e-folding decay time of $20\pm2$ months was obtained from the ANYSO AOT data set and a quite short decay time of $8\pm1$ months in the case of PNE (2017). If we remove the impact of the stratospheric background aerosol and Ulawun volcanic sulfate aerosol (shown in Fig. 6c), the ANYSO AOT decay time decreases to $19\pm1$ months.

Several processes influence the decay time. Besides sedimentation of particles and self-lifting of the absorbing smoke particle, horizontal (meridional) dispersion, and even the growth of the smoke particles by water uptake and condensation of gases on the particles have to be taken into account. Larger particles produce larger optical effects and thus lead to an increase in AOT.

The top height of the smoke layer plays an important role as well. The ANYSO smoke was distributed over the entire lower stratosphere up to heights around 24 km. And this comparably large top height in combination with particle sedimentation was

responsible for the e-folding decay time of 19-20 months as we will further discuss below. The PNE smoke was mostly confined between 14 and 20 km height and thus could be removed quite efficiently (from the stratosphere above 13 km). Furthermore, complex horizontal dispersion features in the Northern Hemisphere have been reported in the case of the Canadian smoke event (Kloss et al., 2019; Pumphrey et al., 2021). The smoke became efficiently distributed towards the tropics as well as towards the North Pole within a few weeks. On the other hand, a clear detection of the Canadian smoke was only possible until

January-February 2018 (up to 5-6 months after injection) so that an accurate decay time could not be computed from such a short time series with quite strongly varying AOT values.

The short decay time for the High Arctic smoke (SILBE, 2019-2020) of 5 months is related to the specific polar meteorological conditions (impact of the strong polar vortex) and specific vertical air mass exchange mechanisms (related to the Brewer-Dobson circulation), and also to the fact that we considered the AOT from the layer base in the upper troposphere

to the top at around 18 km height in the computation. Thus, even removal processes in the upper troposphere (e.g., by cirrus evolution and removal of smoke particles by falling ice crystals) had an impact on the decay behavior.

Table 3 summarized the key features for the three events and allows further comparison. It is worthwhile to compare the ANYSO decay time also with decay times of volcanic perturbations found after the Pinatubo eruption in 1991 at mid-latiutde lidar stations in the Northern and Southern Hemisphere. The Pinatubo aerosol layer also reached to about 25 km height. In

the case of volcanic aerosol, the dominating removal process is sedimentation. Self-lifting and particle growth aspects can be ignored in the analysis of the decay of a volcanic perturbation 1-5 years after the eruption. For the northern hemisphere, our long-term lidar observations of the Pinatubo sulfate aerosol at Geesthacht (close to Hamburg) at 53.4°N in northern Germany,



yielded decay times of 14-15 months (445 days) (Ansmann et al., 1997). Nagai et al. (2010) reported e-folding decay times of 14 and 16.5 months obtained from lidar observations at Tsukuba (near Tokyo, 36.1°N), Japan, and Lauder (45°S), New

Zealand, respectively. According to Sekiya et al. (2016), the Pinatubo-related decay time was 12-13-months on a global scale. Sekiya et al. (2016) performed global-scale modeling constraint to the lidar observations at Tsukuba and Lauder. The shorter Pinatubo decay times are related to the fact that the volcanic sulfate particles showed a characteristic (effective) radius of 0.4-0.5 $\mu$m (Ansmann et al., 1997) whereas the Australian smoke particles were much smaller with effective radii of 0.2-0.3 $\mu$m (Ansmann et al., 2021a) so that the sedimentation speed was much smaller. Self-lifting processes and particle growth effects

had no impact on the decay behavior a few months after injection, when the AOT was mostly <0.05.

    In Fig. 10c, we finally compare the observed particle depolarization ratios of the three fire events. As can be seen, in the case of Canadian smoke over Europe (PNE, 2017), the depolarization ratio decreased slowly with time. This points to a slow smoke aging process in the dry stratosphere at low relative humidity conditions and low levels of condensable gases. It took several months before the initially irregularly shaped particles became very compact and at the end spherical so that

the depolarization ratio approached values <0.05. The aging process seemed to be much faster in the Southern Hemisphere (ANYSO, 2020). A relatively large amount of water vapor and condensable gases were available in the stratosphere so that the smoke particles became spherical in shape within 4-6 weeks after injection. Khaykin et al. (2020) reported a remarkable increase in the stratospheric abundance of the gaseous combustion products which were injected together with the smoke aerosol. Furthermore, the injected mass of water was estimated to be very high (27±10 Tg, about 3% of the total mass of

stratospheric overworld water vapor in the southern extratropics). Unfortunately, it was difficult to determine trustworthy particle depolarization ratios over Punta Arenas since February 2020. However, as was shown in Fig. 8c, the few 355 nm depolarization ratios in March to June 2020 indicate the comparably fast decrease of the depolarization ratio with time and thus fast particle aging. All depolarization value in Fig. 8c were ≤10% since February 2020.

    In strong contrast to these pyroCb-related smoke events, the Siberian smoke (SILBE, 2019) showed a rather different be-

havior. Rather low depolarization ratio values were found at all after the beginning of the MOSAiC campaign in October 2019 (about 80 days after injection into the UTLS height range over Siberia). Lidar measurements of stratospheric smoke layers over Leipzig, Germany, in August 2019 (originating from the SILBE fires according to backward trajectory analysis) and with CALIOP also revealed low smoke depolarization ratios (Ansmann et al., 2021b). This strong contrast to the ANYSO and PNE depolarization features again points to the fact that different lifting processes took place. PyroCb convection was responsible

for fast lifting of smoke towards stratospheric heights (within less than two hours) in the case of PNE and ANYSO, whereas a slow ascent (within more than two days) caused by self-lifting prevailed in the case of SILBE smoke.

## 4   Smoke impact on ozone depletion

Two events of record-breaking ozone depletion were observed in 2020. The first ozone hole was detected over the High Arctic (March-April 2020) (DeLand et al., 2020; Wohltmann et al., 2020; Inness et al., 2020; Wilka et al., 2021; Ohneiser et al., 2021)

and later on, in September-November 2020, another ozone hole appeared over Antarctica (Stone et al., 2021). In September-





November of the following year 2021, again strong ozone depletion was recorded over Antarctica (Stone et al., 2021). All three events occurred during smoke-perturbed stratospheric aerosol conditions. The lower stratosphere over the High Arctic was filled with Siberian wildfire smoke between the tropopause and 15-18 km height in early spring of 2020 (Ohneiser et al., 2021), while Australian smoke caused high aerosol pollution levels from the tropopause up to about 24-25 km height during

the southern hemispheric spring seasons of 2020 and 2021.

In this section, we will briefly discuss our aerosol profile observations at Punta Arenas in combination with ozone profile measurements over two Antarctic sites and Lauder, New Zealand, in September-November 2020. The three stations are indicated in Fig. 9. In a follow-up article, exclusively focusing on the smoke-induced ozone depletion, we will deepen our discussion on the potential role of smoke particles on ozone depletion in the Arctic and Antarctica based on our aerosol obser-

vations during the MOSAiC and DACAPOL-PESO campaign and NDACC ozone profile observations. It is well known that strong ozone depletion requires the development of a cold and long-lasting polar vortex, which favors increased PSC formation. These stratospheric clouds as well as the background aerosol conditions allow for the activation of halogen components via heterogeneous chemical reactions on the surface of these stratospheric particles (in PSCs to 90% on liquid supercooled ternary solution (STS) particles, i.e., on $H_2SO_4/H_2O/HNO_3$ droplets) (Carslaw et al., 1994; Kawa et al., 1997; Wegner et al., 2012;

Kirner et al., 2015). The activated halogen species subsequently destroy ozone molecules in the spring season upon sun exposure. It is also well-known that volcanic sulfate particles can significantly influence these processes by increasing the aerosol particle surface area available for the activation of ozone-destroying halogen components inside and outside of PSCs (Hofmann and Solomon, 1989; Portmann et al., 1996; Dhomse et al., 2015). We documented the impact of the Pinatubo aerosol on ozone depletion based on lidar and ozonesonde observations at Leipzig and Lindenberg (53°N), Germany, respectively, in 1992 and

1993 and found an ozone loss of up to 30% in the strongly polluted lower stratospheric environment (15-20 km height range) where particle surface area concentrations of 25-35 $\mu m^2$ $cm^{-3}$ were present (Ansmann et al., 1996).

Zhu et al. (2018) showed in the case of a minor eruption of the Calbuco volcano in Chile in 2015, producing a 500 nm AOT of less than 0.007 (Bègue et al., 2017), how volcanic sulfate aerosol influences PSC formation, a key aspect leading to large ozone depletion within the polar vortex region. The Calbuco aerosol produced a record-breaking ozone hole in 2015 (compared

to foregoing observations). Column ozone deviations (from the long term mean) as measured over the two selected Antarctic and the New Zealand stations are shown in Fig. 11.

The ozone hole observed in September-November 2020 was two times larger than the Calbuco related ozone hole. It is reasonable to assume that wildfire smoke particles may also be able to contribute to halogen activation and ozone depletion (Yu et al., 2021; Ohneiser et al., 2021; Rieger et al., 2021). However, in contrast to background and volcanic sulfate particles

($H_2SO_4/H_2O$ droplets), the knowledge about the chemical and microphysical properties of aged wildfire smoke (after many months of traveling around the globe) and about the pathways of the smoke impact on ozone depletion is rather limited. We assume that the aged stratospheric wildfire smoke particles are most likely glassy, show a core-shell morphology, and are largely composed of organic material (organic carbon, OC, in the shell) and, to a minor part, of black carbon (BC, concentrated in the core part) (China et al., 2013). Further thin coatings may exist as a result of condensation of trace gases (injected together

with the smoke particles into the stratosphere). The size distribution of the stratospheric smoke particles is well described by





an accumulation mode with effective radius of 0.2-0.3 $\mu$m (Haarig et al., 2018; Ansmann et al., 2021a; Engelmann et al., 2021; Ohneiser et al., 2021). We hypothesize that these glassy accumulation-mode particles may act as condensation sinks for stratospheric $H_2O$, $H_2SO_4$, and $HNO_3$ and, then may perturb the stratospheric background aerosol as well as PSC formation conditions, in a similar way as demonstrated by (Zhu et al., 2018) for volcanic sulfate aerosol.

The injection of wildfire smoke into the UTLS regime with the consequence of severe ozone depletion is a rather new aspect in atmospheric sciences and needs to be explored in future research projects (laboratory studies, airborne in-situ observations, modeling efforts). To initiate and stimulate the required scientific discussion, we showed smoke and ozone profiles over the North Pole region measured during the MOSAiC expedition in the winter and spring seasons of 2020 (Ohneiser et al., 2021; Voosen, 2021). In this section, we will add new observational data to this discussion. The findings in the Southern Hemisphere

are clearer than the High Arctic results.

Figure 11 shows the variations of the ozone partial pressure from the long-term mean. Ozonesonde observations at Lauder (45°S, New Zealand, outside of the polar vortex), the Neumayer station (70.6°S, Antarctica, edge of the polar vortex), and at the South Pole (Antarctica, inside polar vortex) are averaged to determine a robust time series of ozone variations over the last twenty years. (Stone et al., 2021) mentioned the pronounced ozone minima in 2015, 2018 and 2020. However, the 2020-2021

minimum is by far the strongest and longest. In 2019 and early 2020, the ozone deviations were positive and then started to decrease continuously towards the record-breaking negative deviations found between August 2020 and March 2021. The grey area in Fig. 11 highlights this remarkable ozone destruction period. The negative standard deviation was twice as large as earlier minima since 2002.

The other grey period in Fig. 11 highlights the potential impact of the so-called Black Saturday fires between 7 February –

14 March 2009 (Siddaway and Petelina, 2011) on ozone reduction. On Saturday, the 7 February 2009 (coinciding with the left boundary of the 2009 grey column), extraordinarily strong fires in combination with the development of a few pyroCb cloud towers northeast of Melbourne initiated the injection of wildfire smoke into the stratosphere (de Laat et al., 2012). However, in 2009, a smoke-related ozone depletion did not occur. Instead, even a moderate increase of the ozone concentration was observed. As discussed by Peterson et al. (2021), the injected smoke amount was an order of magnitude smaller than the

injected one observed in December 2019 and January 2020. This 2009 smoke event was obviously too weak to have any influence on ozone depletion.

Fig. 12 shows the deviation of the ozone partial pressure (from the 2000-2020 mean value) over the Neumayer research station of the Alfred Wegener Institute (70.6°S, 8.3°W). The daily smoke layer base and top heights as measured over Punta Arenas (53.2°S, 70.9°W) are indicated by grey and black circles, respectively. As mentioned above, data gaps in the lidar time

series are caused by cloudy weather or instrumental problems. We assume that the general structures of the smoke layers as measured over Punta Arenas were similar to the ones over the Neumayer station (about 1900 km further south). In agreement with Fig. 11, large positive ozone deviations were observed in November and December 2019, before the long-lasting decrease of the ozone partial pressure over about 10 months started. From October 2020 to January 2021 the largest negative ozone deviations were found between 12 and 25 km, well within the Australian smoke layer. As in the case of the strong ozone

depletion over the High Arctic (Ohneiser et al., 2021), the polar vortex was unusually strong over Antarctica in July and





August 2020 (Copernicus, 2021). The large-scale warming of the Southern Hemispheric stratosphere (Stocker et al., 2021) as a result of strong absorption of solar radiation by smoke in Januray to April 2020 may have contributed to the development of a strong vortex over Antarctica in the July-September 2020 by suppressing large-scale wave activity (Tritscher et al., 2021).

Figure 13 shows height profiles of the September-November 2020 mean particle surface area (SA) concentration together with September-November mean ozone deviations for the long-term mean and the height range with PSC as observed with CALIOP. The 2000-2020 mean ozone profile includes again the observations at Lauder (New Zealand), Neumayer station (Antarctica), and the South Pole station (Antarctica) (NDACC, 2021). The PSC height profile was retrieved by using the CALIPSO V4 classification scheme (CALIPSO, 2021a). All CALIOP data for the Southern Hemisphere during the winter season 2020 were downloaded and the number of PSC entries obtained with the CALIPSO V4 classification were available as a function of height. The maximum PSC number was then arbitrarily set to 0.95 in our value range from 0 to 1 in Fig. 13b. Below 13 km height cirrus clouds are frequently misclassified as PSC, therefore the PSC height range is shown down to 13 km (in arbitrary units) only. The microphysical properties of the smoke (SA, $n_{50}$, $n_{250}$ in Fig. 13a) were obtained by converting the September-November 2020 mean smoke extinction coefficient profile by using the conversion scheme of Ansmann et al. (2021a). $n_{50}$ (accumulation mode) and $n_{250}$ (large particle fraction) considering particles with radius >50 nm and >250 nm, respectively. The background extinction coefficient profiles in Sakai et al. (2016) were converted into SA, $n_{50}$, and $n_{250}$ values as well (dashed-dotted grey line in Fig. 13a) by using conversion factors for typical sulfat background accumulation-mode particles with an effective radius around 0.15 $\mu$m. Effective radii of aged smoke particles are between 0.2 and 0.3 $\mu$m (Ansmann et al., 2021a).

We observed a clear coincidence of the layer with strongest ozone reduction over the ozonesonde stations (14-25 km height range) with the layer showing an enhanced particle surface-area concentration over Punta Arenas (10-24 km height range), and the height range in which CALIOP detected PSCs (mostly over Antarctica at heights from 13-26 km). As can be seen in Fig. 13a, the particle surface area concentration was about a factor 2, 3, and 5 larger at 22, 20, and 15 km height than at unperturbed conditions, respectively. Surface area values of 0.5-5 $\mu$m$^2$ cm$^{-3}$ and particle number concentrations of 25-65 cm$^{-3}$ (for $n_{50}$) and 0.15-1 cm$^{-3}$ or 150-1000 particles per liter (for $n_{250}$) in the height range from 15-22 km, in which the highest values of ozone reduction were found, clearly indicate a significantly polluted lower stratosphere. The large particle fraction could be regarded as the reservoir of most favorable ice-nucleating smoke particles available for PSC ice formation processes.

According to the dashed-dotted line in Fig. 13a clean background conditions are characterized by values of 2-7 cm$^{-3}$ for $n_{50}$ and 0.05-0.2 cm$^{-3}$ for $n_{250}$. The surface area concentration of 0.3-1 $\mu$m$^2$ cm$^{-3}$ from 15–22 km height (according to the dashed line) is in good agreement with balloon-borne observations over Laramie (41°N), Wyoming, of 0.5-1 $\mu$m$^2$ cm$^{-3}$ at 18-20 km height during volcanic quiescent time in the late 1970s (Hofmann and Solomon, 1989) and in the late 1990s (Deshler et al., 2003).

We assume that in the PSC height range from 13-26 km all background particles and most of the H$_2$SO$_4$/H$_2$O-coated smoke particle grew by HNO$_3$ condensation during PSC events so that the particle surface area concentration and thus the potential for halogen activation increased by an order of magnitude within the PSCs (Carslaw et al., 1994; Tritscher et al., 2021). The



total particle number concentration $n_{\text{tot}}$ is typically a factor of 4-6 higher than $n_{50}$ (Zhu et al., 2018). In the case of the Calbuco volcanic impact, there were around 100 volcanic particles per cm$^3$ in the height range of observed high ozone depletion and respectively about 100 liquid PSC particles per cm$^3$ (with a size distribution showing a small effective radius below 0.1 $\mu$m) (Zhu et al., 2018). The numerous, small PSC particles were probably of key importance for the unusually large ozone depletion in September 2015 (after the Cabuco eruption in April 2015). Similarly, for our reported case, high numbers of smoke particles concentrations (probably $n_{\text{tot}} > 100$ cm$^{-3}$) were found in the PSC height region during the Southern Hemispheric winter and spring months in 2020 and were probably involved in PSC formation, strong halogen activation and the resulting record-breaking ozone depletion.

## 5 Conclusion and outlook

We presented a detailed characteriziation of the geometrical, optical, and microphyscial properties of the largest stratospheric wildfire-smoke-related stratospheric perturbation ever observed around the globe. The Australian smoke sensitively affected the radiation budget, dynamical processes, and most probably ozone depletion in the stratosphere of the Southern Hemisphere. We documented this unique and record-breaking wildfire pollution event with two ground-based lidars in southern South America in 2020 and 2021 and found a slow decrease of the smoke perturbation until the end of 2021. Our height-resolved observations can be regarded as representative for mid and high latitudes in the Southern Hemisphere. The presented observations can be regarded as a benchmark data set for modeling studies and for comparison with active and passive remote sensing products in the framework of quality assurance efforts.

For the second time, a record-breaking ozone hole developed in 2020 in a smoke-polluted stratospheric environment. In the case of the unusually strong Arctic ozone depletion (in March 2020), the impact of smoke on ozone reduction mainly occurred in the lowest part of the stratosphere (10-17 km height range) and thus below the region with strongest ozone depletion (Ohneiser et al., 2021). This time, the height range with pronounced ozone depletion over Antarctica (September-November 2020) fully overlapped with the Australian smoke layer. Strong ozone depletion over Antarctica was even observed in September-November 2021, again at significantly enhanced smoke pollution levels. We will report on this in a follow-up article. As we already hypothesized in Ohneiser et al. (2020), if upcoming studies indicate a link between huge fires (caused by unusually hot temperatures and droughts as a result of climate change), corresponding smoke occurrence in the lower stratosphere, and severe ozone depletion in the Arctic and Antarctica, the climate change debate will be added by a new, and until now, not considered important aspect. In other words, climate change will not only change stratospheric chemistry and dynamics by causing stratospheric temperatures to decrease associated with a warming troposphere but also by injection of new types of particles, commonly only present in the troposphere. A new parameter we have to consider in future climate and ozone depletion predictions.

Another research topic is the impact of smoke on cirrus formation in the upper troposphere. Unique data sets for the High Arctic (October 2019 to May 2020) and over southern South America (January 2020 to November 2021) are available for in-depth smoke-cirrus interaction by means of the full active remote sensing equipment available during the MOSAiC as well

665 as during the DACAPO-PESO long-term campaigns. The tropopause region was permanently polluted over Punta Arenas in
2020 and 2021. A first case study of the smoke impact on cirrus formation was presented by Engelmann et al. (2021) observed
during the MOSAiC expedition.

## 6 Data availability

Polly lidar observations (level 0 data, measured signals) are in the PollyNet database (Polly, 2021). LACROS observations
(level 0 data) are stored in the Cloudnet database Lacros (2021). All the analysis products are available at TROPOS upon
request (info@tropos.de). CORAL data is available at TROPOS and DLR, Oberpfaffenhofen, upon request. The ozone data is
used from the regular ozone launch data at Lauder (New Zealand), Neumayer Station (Antarctica), and South Pole (Antarctica)
available at the Network for the Detection of Atmospheric Composition Change (NDACC) website (NDACC, 2021). Besides
the continuous lidar measurements, an Aerosol Robotic Network (AERONET) sunphotometer was operated (AERONET,
2021). Fire and MODIS data are available at the NASA data base (MODIS, 2021a, b, c; FIRMS, 2021). CALIPSO observations
were downloaded from the CALIPSO data base (CALIPSO, 2021a, b, c). The ozonesonde data can be found by using the
link NDACC (2021). The radiosonde data are available at https://doi.org/10.1594/PANGAEA.928656 (Uni-Wyoming, 2021).
Sentinel-5 aerosol data is available at (Sentinel-5, 2021). GDAS1 (Global Data Assimilation System 1) re-analysis products
from the National Weather Service's National Centers for Environmental Prediction are available at (GDAS, 2021).

## 7 Author contributions

The paper was written by KO and AA, and designed by KO, AA, AC, and DAK. The data analysis was performed by KO,
AC, and DV supported by AA, HB, PS, and RE. BK and NK prepared and provided CORAL data and supported the data
analysis. BB, CJ, MR, and FZ were involved in the DACAPO-PESO campaign and took care of the station and the continuous
measurements. All co-authors contributed to the discussion of the results.

## 8 Competing interests

The authors declare that they have no conflict of interest.

## 9 Financial support

The authors acknowledge support through the European Research Infrastructure for the observation of Aerosol, Clouds and
Trace Gases ACTRIS under grant agreement no. 654109 and 739530 from the European Union's Horizon 2020 research and
innovation programme. We thank AERONET-Europe for providing an excellent calibration service. AERONET-Europe is
part of the ACTRIS project. The field observations at Punta Arenas were partly funded by the German Science Foundation
(DFG) project PICNICC with project number 408008112. This research has been supported by the U.S. Department of Energy





(grant no. DE-SC0021034). The Polarstern Polly data was produced as part of the international Multidisciplinary drifting Observatory for the Study of the Arctic Climate (MOSAiC) with the tag MOSAiC20192020 and Project ID AWI_PS122_00. The development of algorithm for inversion of lidar observations was supported by Russian Science Foundation (project 21-17-00114).

*Acknowledgements.* We thank AERONET for their continuous efforts in providing high-quality measurements and products. We also thank the CALIPSO team for their well-organized easy-to-use internet platforms. We are grateful to the NDACC ozonesonde teams at Neumayer station, Lauder, and South Pole for performing high quality ozone profile observations.



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





**Table 1.** Layer-mean particle optical (532 nm) properties and estimated microphysical properties of the rotating and ascending smoke disc on 29 January 2020, when the center of the vortex was close to Punta Arenas (Ohneiser et al., 2020; Ansmann et al., 2021a). Besides values obtained by conversion of extinction coefficients into microphyscial properties (Ansmann et al., 2021a), the lidar inversion method was applied to Polly multiwavelength backscatter and extinction observartions on 26 January (Veselovskii et al., 2002) to add values for the effective radius, refractive index, and single scattering albedo. Uncertainties are of the order of 15-30% in the optical properties, 20-50% in the microphysical properties. $\pm 0.1$ for the real part of the refractive index, within an order of magnitude for the imaginary part, and $\pm 0.05$ for the single scattering albedo. The number concentration $n_{50}$ considers particles with radius $>50$ nm only.

| Parameter | 29 January 2020 |
|---|---|
| Height, base to top [km] | 20.5-26.0 |
| Backscatter coefficient $\beta_{532}$ [Mm$^{-1}$ sr$^{-1}$] | 0.7 |
| Extinction coefficient $\sigma_{532}$ [Mm$^{-1}$] | 75 |
| Lidar ratio $L_{532}$ [sr] | 106 |
| AOT $\tau_{532}$ | 0.31 |
| Linear depolarization ratio $\delta_{532}$ | 0.21 |
| Number concentration $n_{50}$ [cm$^{-3}$] | 500 |
| Effective radius $r_{\text{eff}}$ [$\mu$m] | 0.29 |
| Volume concentration $V$ [$\mu$m$^3$ cm$^{-3}$] | 8.7 |
| Mass concentration $m$ [$\mu$g m$^{-3}$] | 10 |
| Surface area concentration $s$ [$\mu$m$^2$ cm$^{-3}$] | 130.0 |
| Refractive index, real part | 1.48 |
| Refractive index, imag. part | 0.02 |
| Single scattering albedo, 532 nm | 0.79 |





**Table 2.** Mean values and standard deviations of the Ångström exponents (backscatter-related, $A_\beta$, lidar ratio related, $A_{LR}$, and extinction related, $A_\sigma$) and lidar ratios (LR), computed from all stratospheric values shown in Fig 8. Lidar ratio index denotes: A: Aeronet-Polly retrieval, P: Polly observation, C: CORAL observation, all: mean values by considering all determined and retrieved LR values. Further explanation is given in the text.

| Parameter | Average | Standard deviation |
|---|---|---|
| $A_{\beta,355,532}$ | 1.32 | 0.79 |
| $A_{\beta,355,1064}$ | 1.22 | 0.97 |
| $A_{\beta,532,1064}$ | 1.70 | 0.59 |
| $A_{LR,355,532}$ | -0.68 | |
| $A_{\sigma,355,532}$ | 0.64 | |
| $LR_{P,355}$ | 71 sr | 14 sr |
| $LR_{A,355}$ | 67 sr | 22 sr |
| $LR_{P,532}$ | 95 sr | 14 sr |
| $LR_{C,532}$ | 82 sr | 9 sr |
| $LR_{A,532}$ | 93 sr | 19 sr |
| $LR_{A,1064}$ | 120 sr | 21 sr |
| $LR_{all,355}$ | 69 sr | 19 sr |
| $LR_{all,532}$ | 91 sr | 17 sr |
| $LR_{all,1064}$ | 120 sr | 22 sr |

**Table 3.** Comparison of the three wildfire smoke events (ANYSO: Australian fires, PNE: Canadian fires, SILBE: Siberian fires) in terms of mean values and SD of smoke layer geometrical and optical properties and perturbation decay time. The observations were performed over Europe (PNE), Punta Arenas (ANYSO), and the North Pole region (SILBE). For these sites and regions (see Fig. 9) the mean tropopause height and SD and, for Europe, the typical tropopause height range are given as well.

| Parameter | ANYSO | PNE | SILBE |
|---|---|---|---|
| Injected mass [Tg] | 0.8-1.1 | 0.1-0.3 | − |
| Tropopause height [km] | 11.1±1.4 | 10-13.5 | 9.1±1.0 |
| Layer base height [km] | 9.6±1.6 | 16.2±2.8 | 7.9±1.2 |
| Layer top height [km] | 22.6±1.7 | 18.1±2.5 | 16.2±1.4 |
| Layer depth [km] | 12.9±2.3 | 1.9±1.5 | 8.5±1.7 |
| AOT (day 80-100) | 0.034 | 0.007 | 0.06 |
| Layer mean particle ext. cf. [$Mm^{-1}$] | 3.2±1.7 | 1.5±1.4 | 5.2±2.6 |
| Decay time [months] | 20±1 | 8±1 | 5±0 |



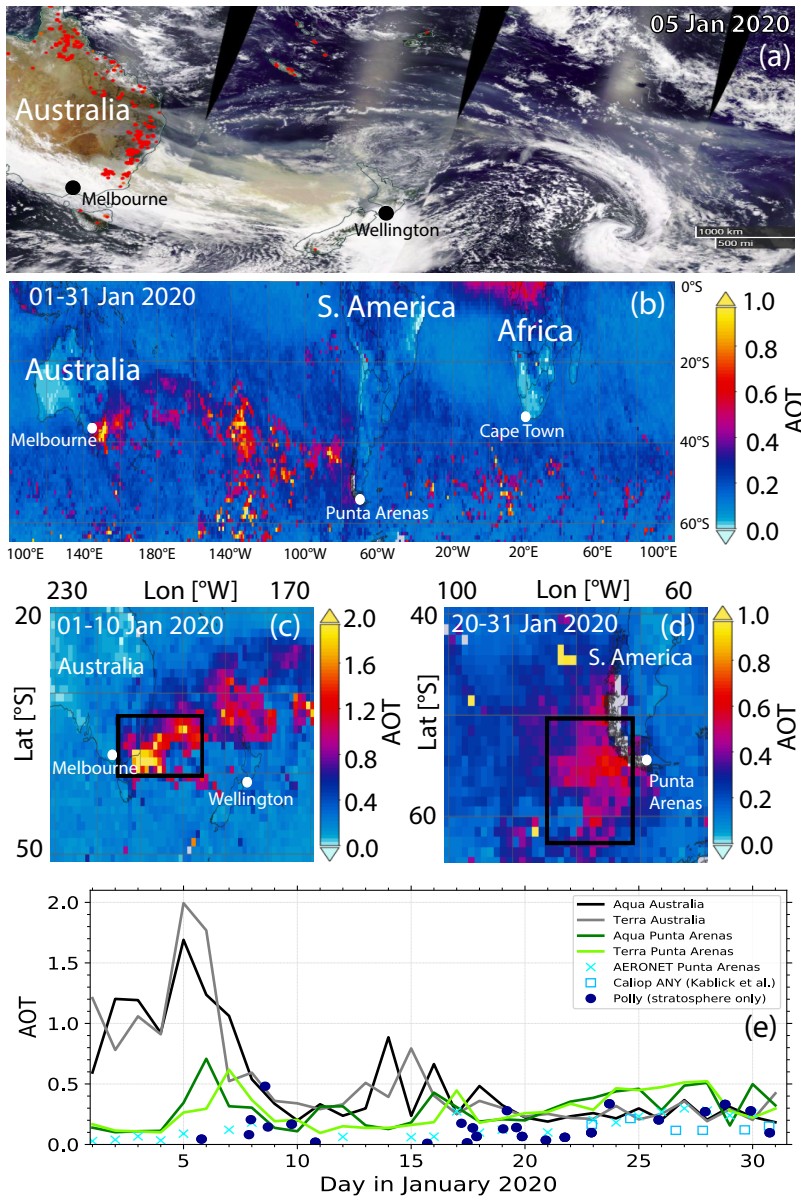

**Figure 1.** Satellite observations of the stratospheric smoke transport from Australia to South America in January 2020: (a) Brownish smoke layer between Melbourne and Wellington on 5 January 2020 (MODIS, 2021a, b) and fires indicated by red spots in eastern and northern Australia (FIRMS, 2021). (b) 500 nm AOT in the southern hemisphere (MODIS, 2021c), the smoke plumes were traveling mostly eastward. (c) Mean AOT values for the 1-10 January period, close to the source region (MODIS, 2021c). (d) Mean AOT values for the 20-31 January period, close to southern South America (MODIS, 2021c). (e) Time series of daily 500 nm AOT (mean values for the black rectangles shown in c and d separately for MODIS Terra and Aqua. Cloud screened values are shown (MODIS, 2021c). In addition, AERONET sunphotometer values of the total (tropospheric+stratospheric) AOT at Punta Arenas, CALIOP-derived AOT values as shown in Kablick et al. (2020), and Polly AOT values for the stratosphere over Punta Arenas, are shown.





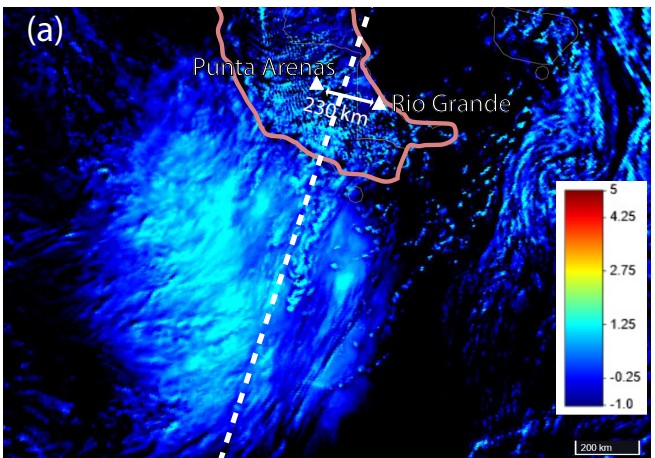

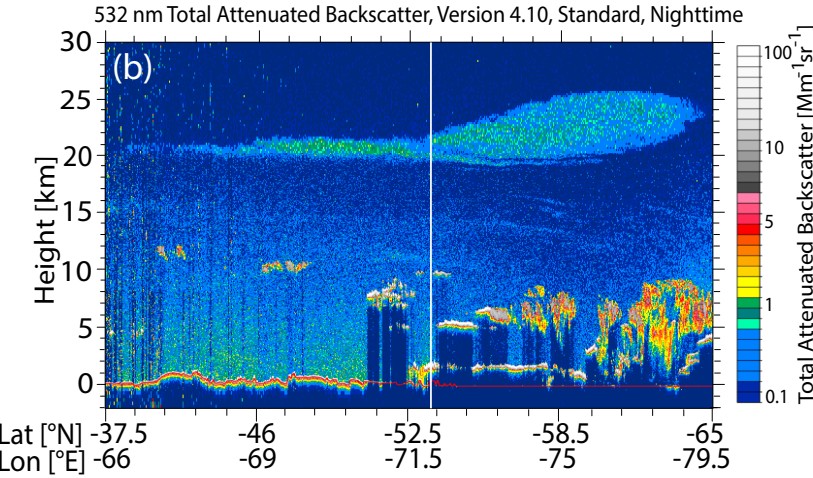

**Figure 2.** (a) Sentinel-5 aerosol index (340-380 nm), showing the location of the rotating and ascending smoke vortex southwest of the southernmost tip of South America on 26 January 2020, 00:00 UTC-20:00 UTC (Sentinel-5, 2021). Indicated are the two ground-based lidar stations at Punta Arenas and Río Grande (white triangles). The white dashed line indicates the CALIIOP overpass. (b) CALIOP observation along the overpass track on 25 Jan 2020 at 05:50 to 06:02 UTC showing smoke plumes above 20 km and the smoke-filled vortex at latitudes south of 52.5°S in terms of the 532 nm attenuated backscatter coefficient. The thin vertical line at 52.3° indicates the shortest distance to the two ground-based lidars (triangles in a).





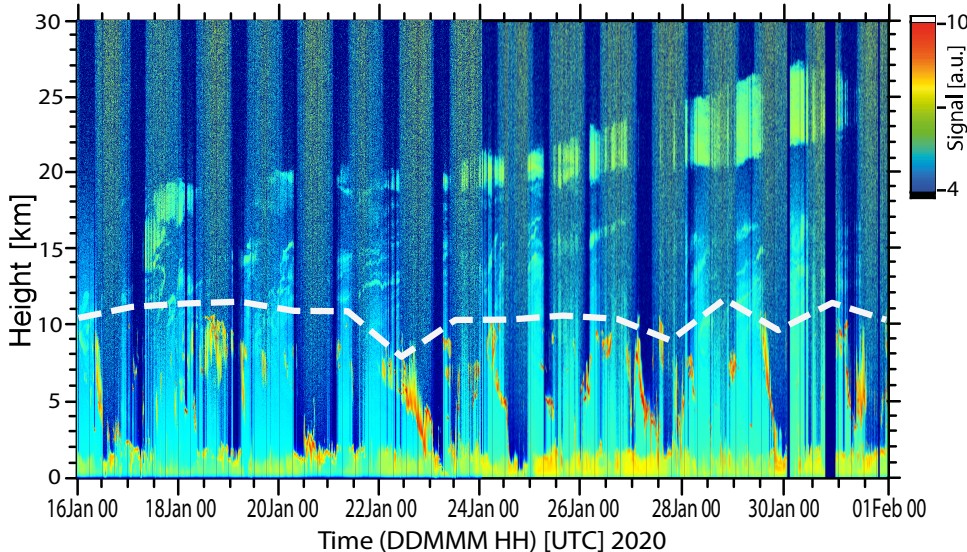

**Figure 3.** The rotating and ascending smoke vortex as observed with ground-based lidar at Punta Arenas from 24-31 January 2020. Shown is the range-corrected signal at 1064 nm wavelength in arbitrary units in logarithmic color scale. The tropopause is indicated by the white dashed line. Tropospheric clouds and day and night signal background changes cause the many interruptions of high quality observations of the stratospheric smoke.





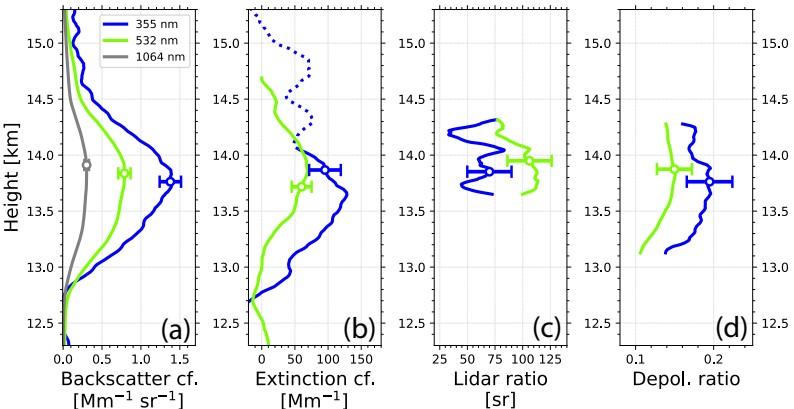

**Figure 4.** Profiles of smoke optical properties measured with Polly over Punta Arenas on 8 January 2020 06:45-07:42 UTC. (a) Particle backscatter coefficient at three wavelengths, (b) smoke extinction coefficient, (c) smoke extinction-to-backscatter ratio (lidar ratio), (d) particle linear depolarization ratio. Error bars show the estimated uncertainties. Profiles of backscatter (in a) and depolarization ratio (in d) were vertically smoothed with 500-700 m signal smooting length and the profiles of extinction coefficient (in b) and lidar ratio (in c) are obtained with a regression window length of 2 km. The dotted part of the 355 nm extinction profile indicates that the basic Raman lidar signals were too noisy to allow an accurate determination of the 355 nm extinction coefficient at heights above the layer center. In the case of the extinction coefficient and lidar ratio profiles signal smoothing was 2000–2500 m.

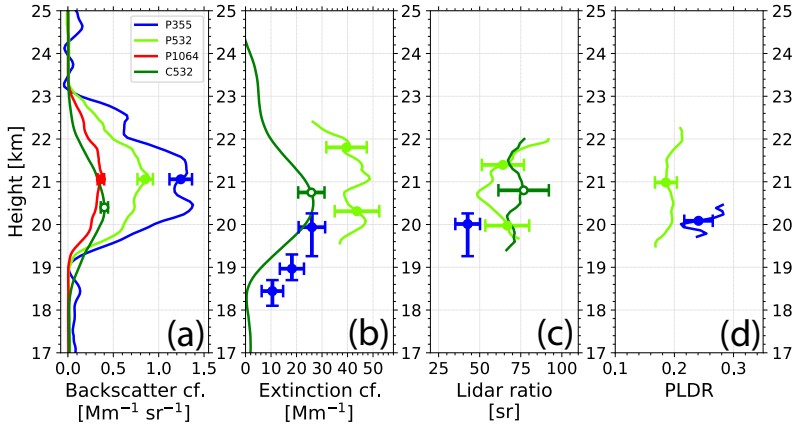

**Figure 5.** Profiles of smoke optical properties measured with Polly (P, closed circles) over Punta Arenas and and with CORAL (C, open circles) over Río Grande on 26 January 2020 between 04:27-06:18 UTC (Polly) and 6:14-8:02 UTC (CORAL). (a) Particle backscatter coefficient at three wavelengths, (b) smoke extinction coefficient, (c) smoke extinction-to-backscatter ratio (lidar ratio), (d) particle linear depolarization ratio. Error bars show the estimated uncertainties. The extinction coefficient and lidar ratio profiles at 355 nm wavelength were noisy, so only a few trustworthy values are shown. Smoothing and regression window lengths are 500 m (Polly, backscatter, depolarization) and 2 km (extinction, lidar ratio). 2 km smoothing and regression window lengths are used in the case of the CORAL profiles.

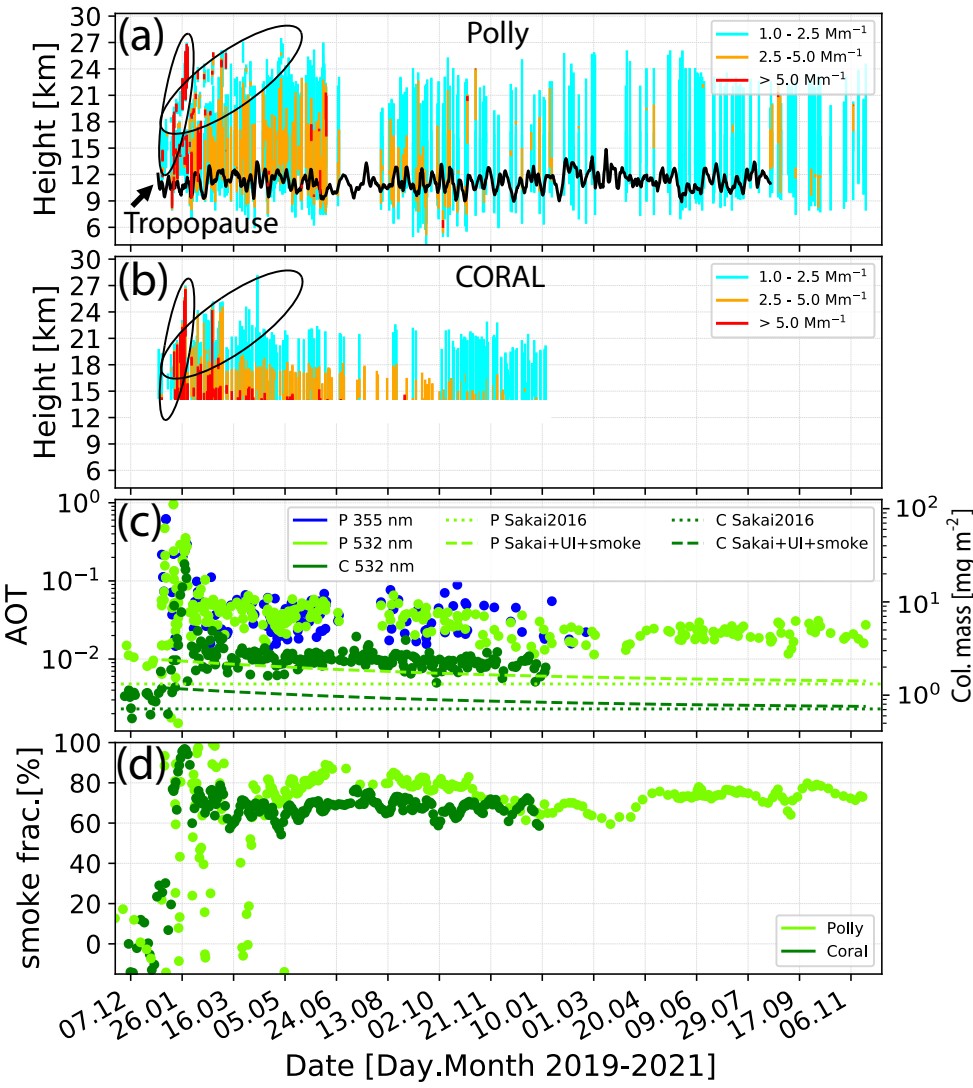

**Figure 6.** (a) Overview of Polly observations of the UTLS smoke layers (colored bars from layer base to top, one bar per day) from 15 November 2019 to 15 November 2021. Observational gaps between bars are caused by periods with opaque clouds or instrumental problems. The colors in each bar indicate height segments with different extinction coefficient levels, see legend in the panel). Black oval circles highlight periods with frequently detected ascending layers. (b) same as (a) except for CORAL observations with minimum measurement height of 14 km. (c) Smoke layer AOT (C denotes CORAL, P denotes Polly) at 355 nm and 532 nm, calculated from the daily profiles of the particle backscatter coefficient, multiplied by a lidar ratio of 55 sr (355 nm) and 85 sr (532 nm). The Polly instrument measures the total UTLS smoke AOT and the CORAL system measures the AOT between 15 and 28 km. (d) Smoke contribution to total AOT which includes contributions by the stratospheric background aerosol (dotted lines, P Sakai2016, C Sakai2016), by Ulawun volcanic aerosol (Ul), and by Australian smoke reaching the stratosphere from September to December 2019 (smoke). The Ulawun and 2019 smoke contributions decreased with time (dashed curves, see text for more details). Fluctuations in the smoke fraction around 0 visible until May 2020 are caused by measurement noise.





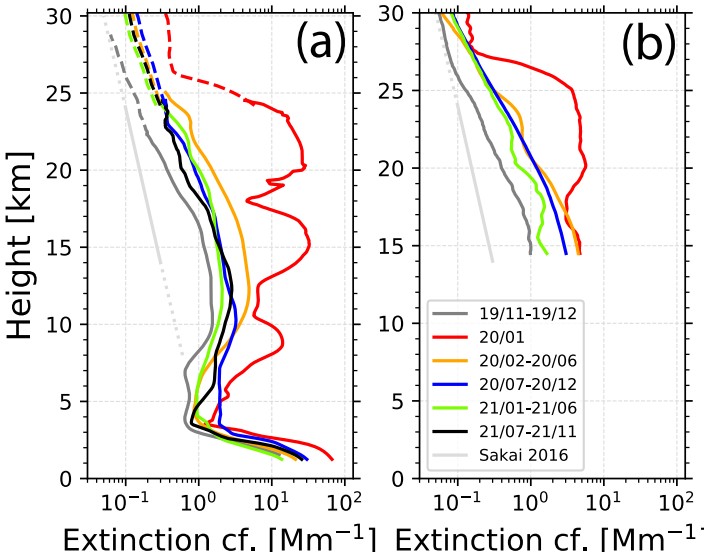

**Figure 7.** January 2020 mean (red), February-June 2020 mean (orange), and half-year mean profiles of smoke extinction coefficient at 532 nm: (a) Polly observation (1-30 km) and (b) CORAL (15-30 km). The particle extinction coefficients were retrieved by multiplying the backscatter coefficient profiles by 85 sr. All solid profiles are average profiles calculated with the Raman method. The dashed lines in (a) were were calculated with the Fernald method (as explained in Sect. 2.1). The thin solid-dotted grey lines show stratospheric background values (Sakai et al., 2016). The November-December-2019 mean extinction profile (before the ANYSO event) is given as dark grey curve, here a lidar ratio of 60 sr is used to convert backscatter into extinction values.





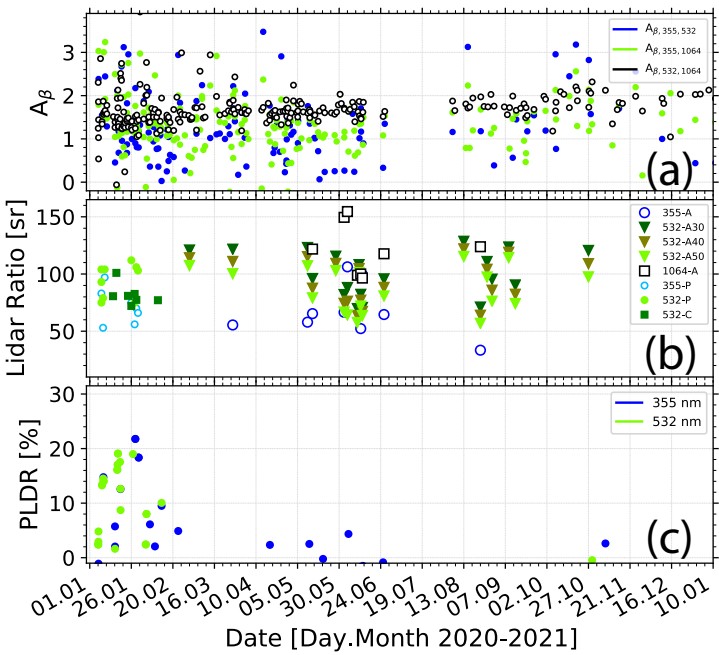

**Figure 8.** Smoke layer mean optical properties observed from 1 January 2020 to 10 January 2021: (a) Backscatter-related Ångström exponent $A_\beta$, (b) extinction-to-backscatter ratio (lidar ratio) from Polly (P) and CORAL (C) measurements in January-February 2020, and from combined lidar-AERONET observations (since March 2020) by using lidar ratios of 30 (A30), 40 (A, A40), and 50 sr (A50) in the estimation of the tropospheric 532 nm AOT from lidar observations (see further explanation in the text). Stratospheric lidar ratios at 355 nm (circles) and 1064 nm (squares) are shown for a lidar ratio of 40 sr in the tropospheric AOT retrieval. (c) Particle linear depolarization ratio (PLDR). The scatter in the data is mainly caused by signal noise and to a minor part by atmospheric variability.

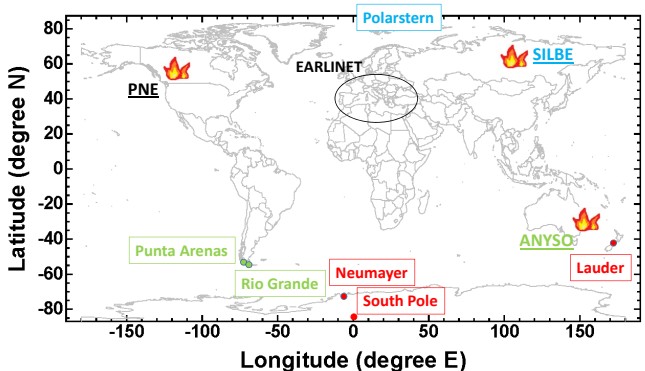

**Figure 9.** Map showing the ground-based lidar stations at Punta Arenas (ANYSO smoke observations in 2020/21), in the High Arctic (Polarstern, SILBE smoke observations in 2019/20), in Europe (PNE smoke observations in 2017/18). In addition, the ozonesonde stations at Lauder, New Zealand, at the Neumayer station, and at the South Pole are indicated.



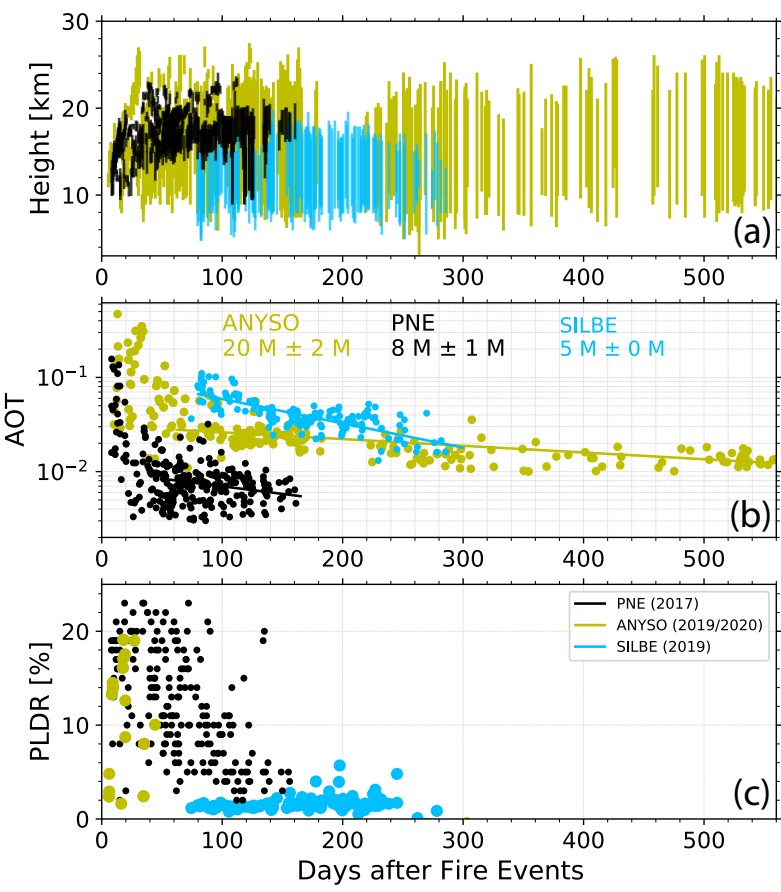

**Figure 10.** Comparison of geometrical and optical properties of three stratospheric fire smoke events measured with different Polly instruments. Day 0 is the 12 August 2017 (PNE), 23 July 2019 (SILBE), and 31 December 2019 (ANYSO). Canadian (PNE) and Siberian (SILBE) fire smoke data are taken from Baars et al. (2019) and Ohneiser et al. (2021), respectively. (a) Overview of Polly observations of UTLS smoke layers (colored bars from layer base to top) in days after smoke injection into the UTLS height regime. (b) 532 nm AOT times series for the different wildfire events, straight lines indicate the decay of the perturbations according to the given e-folding decay times in months (M). In the case of PNE and ANYSO observations, AOT for heights >13 km is used. (c) Particle depolarization ratios (layer mean values) for the different fire events.





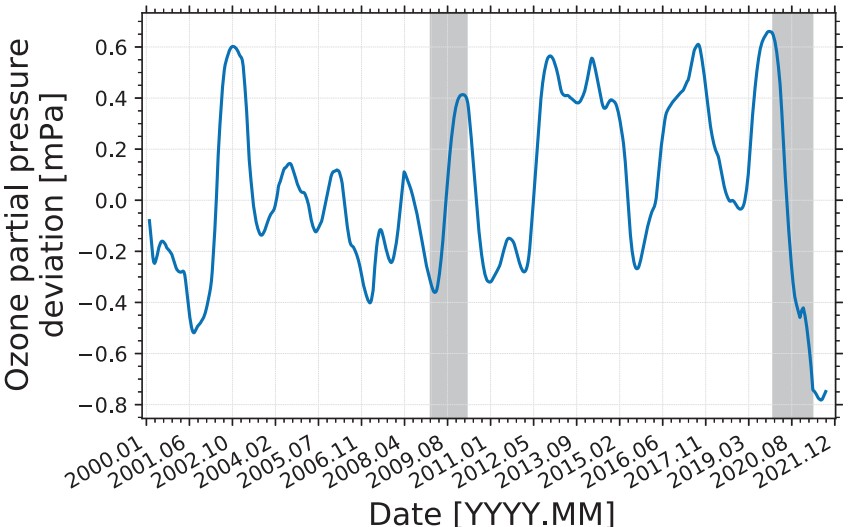

**Figure 11.** Deviation of the vertically averaged ozone partial pressure from the respective 2000-2020 long-term mean ozone value (of 5.9 mPa). Ozone profiles measured regularly (typically one ozonesonde launch per week) at Lauder (New Zealand), the German Antarctic Neumayer Station, and at the South Pole (data from NDACC, 2021) are considered. Grey columns mark two time periods after major Australian fire events in 2009 (Black Saturday) and 2019/2020 (Black Summer).

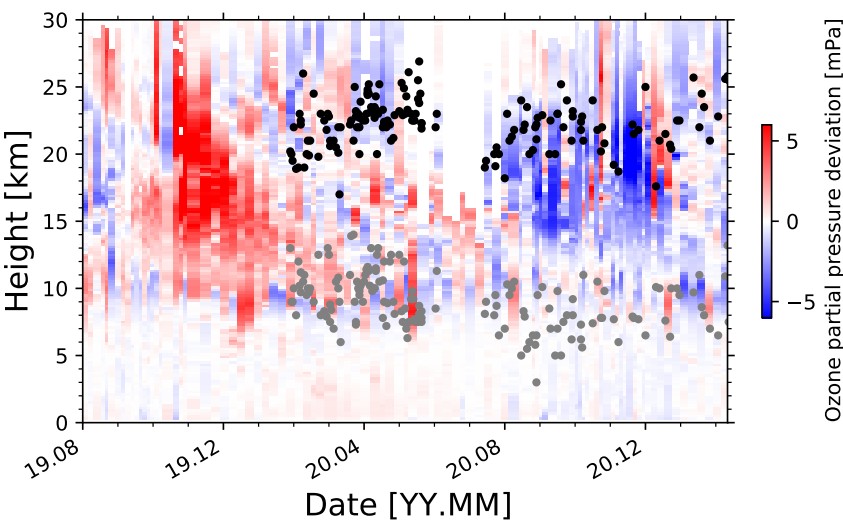

**Figure 12.** Deviation of the ozone partial pressure (with respect to the 2000-2020 mean value) at the Neumayer Station (data from NDACC, 2021). The base (grey dots) and top heights (black dots) of the Australian smoke layer measured with Polly at Punta Arenas indicate the smoke-polluted height range.



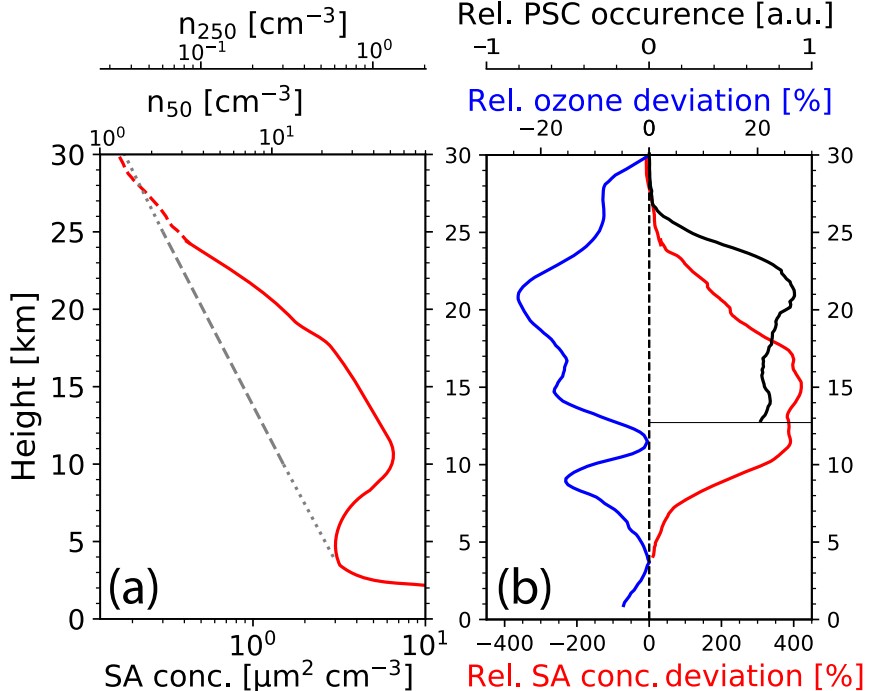

**Figure 13.** (a) Estimated mean profiles of the surface area concentration (SA) and number concentrations $n_{50}$ and $n_{250}$ (particles with radius >50 nm and >250 nm, respectively) for the September-November 2020 time period derived from Polly measurements at Punta Arenas (red). Respective background aerosol estimates are given as a grey dashed line, based on lidar observations at Lauder, New Zealand (Sakai et al., 2016). Dotted line segments indicate extrapolations. (b) Relative increase of the estimated particle surface area concentration (computed by using the grey and red profiles in a) and relative deviation of the ozone partial pressure (mean deviation for the September-November 2020 period) from the 2000-2020 mean ozone value, calculated from the ozonesonde observations at Lauder, New Zealand, the German Antarctic Neumayer Station, and the South Pole station (data from NDACC, 2021). The height range with PSCs in the Southern Hemisphere north of 81.8°S in the winter of 2020 (June-August 2020) above 13 km height (indicated by a horizontal line) obtained from CALIOP observations (CALIPSO, 2021a) is shown in addition. The PSC occurrence frequency per height level is normalized such that the maximum PSC number is 0.95 (further explanations are given in the text).