# Peer review of "Australian wildfire smoke in the stratosphere: the decay phase in 2020/21 and impact on ozone depletion"

_Atmospheric Chemistry and Physics, 2021_

## Author Response (AR1)

Dear Editor, Dear Reviewer 1!

We thank you for careful reading of the manuscript and for providing us with valuable comments to improve the manuscript.

All changes in the revised manuscript are given in **BOLD**

The comments of Reviewer 1 are given in **BLUE**, our answers are given in **GREEN**.

The manuscript presents very interesting long-term observations of the stratospheric wildfire smoke event captured in the Southern Hemisphere. They present the geometrical, optical, and microphysical properties of the smoke layer. The decay behaviour of such a stratospheric perturbation, and the smoke impact on the ozone hole over Antarctica were discussed.

The dataset is interesting, and the manuscript is well written. The manuscript is worthwhile to be published, after addressing all the points raised by reviewers.

Please see below some suggestions and comments:

l17, please specify the wavelength of ext.

Corrected (at 532nm).

l79, does Polly continuously measure the Raman signals?

Yes, it does. We improved the full sentence.

l226, "This is in agreement with the CALIOP observations." please specify.

We now write: This is in agreement with the CALIOP observations shown in \citet{Khaykin2020} who found that the initial ascent rate was at 0.45~km~day$^{-1}$, ......

l234-237, please check and make this sentence clearer.

We improved the explanation and extended the discussion on the findings in Table 1.

l295, "on most of the days", specify the period referred.

Now specified in text. We write: … for the next two years.

l307, the CORAL AOT values are only for the smoke layer?

CORAL minimum measurement height is 15 km! Therefore, we have observations between 15km and smoke layer top. This is now better specified.

l314, "total AOT", please specify the total AOT here.

This is now improved: The contribution of Australian smoke to the total AOT is shown in Fig. 8d. The total AOT includes also contributions by Australian smoke reaching the stratosphere from September-December 2019, Ulawun volcanic aerosol, and background particles.

l535, please define PSC.

Improved.

Table2 does the std here present the time variation? what is the number of profiles used for the calculation? What about provide additional values for the intense period, e.g. only for 1/2020

This is now explained in the Table caption:  The standard deviations include atmospheric variability and dominating retrieval uncertainties. Further explanation is given in the text.

In the text we provide numbers: The statistics consider 366 values for each Angstrom exponent, 7-10 values for each of the lidar ratios, except for the Polly-AERONET 532 nm lidar ratio (19 values). The PLDR statistics are based on 14 (355~nm) and 17 values (532~nm).

The intense period (January 2020) was discussed in Ohneiser et al. (2020).

Fig.1-2 add lat lon in fig.1a and fig.2a

Improved.

Fig7. Maybe provide some uncertainty bars in the figure.

Improved.

Fig. 8, there are negative PLDR values shown in c. Maybe use some thresholds (e.g. SNR or bsc value) to screen out too noisy data.

We checked all cases again (visual inspection) and decided at the end: too noisy signals! So we removed the questionable values.

Dear Editor, Dear Reviewer 2!

We thank you for careful reading of the manuscript and for providing us with valuable comments to improve the manuscript.

All changes in the revised manuscript are given in **BOLD**

The comments of Reviewer 2 are given in BLUE, our answers are given in GREEN.

**General comments**

The article "Australian wildfire smoke in the stratosphere: the decay phase in 2020/2021 and impact on ozone depletion" by K. Ohneiser et al. presents the long time series of lidar observations of the Australian wildfires 2019-2020 on the transport path of the smoke plumes. The location of the lidar site and the occurrence of this record-breaking wildfire make these observations unique and undoubtedly valuable. These 2-year observations contribute to studies of wildfire smoke particles properties, dynamics and their evolution in the atmosphere. This paper also points out the role of aged wildfire smoke in the depletion of ozone, which requires more scientific investigations. The subject fits well the scope of the journal Atmospheric Chemistry and Physics, the manuscript is very well written and the data are well presented. I would suggest this paper being accepted with minor corrections.

**Specific comments:**

L79: elastically … elastic

The full sentence is changed.

L80: 387 and 607 nm include not only nitrogen but also oxygen Raman lines

We changed the text, now: nitrogen Raman channels, water vapor channel

L106-107: Should give approximate definition about fresh and aged smoke

We removed 'fresh and 'aged', too complicated! Aging needs a few days in the troposphere, and a few months in the stratosphere.

L133-135: More information about ozone measurements should be given for non-expert, ex. What is the "column ozone deviation" ?

More details are given now regarding the ozonesonde observations, since when ozone is observed regularly (since the 1990s), and what is measured (ozone partial pressure). This is enough to mention in this section.

L148-149: "We noticed that … cross-talk effect in the Raman channel (less than 0.8%)": this sentence is confusing, what cross-talk, Raman signal contaminated by elastic channel.

We improved the text. Cross talk is caused by elastically backscattered laser photons.

L231: Did you check AERONET inversions to confirm the strong absorption of Australian smoke?

Yes, we checked the AERONET data (already in January 2020). But there are many question marks with AERONET observations of optically thick layers 10-20 km far away. We realized that already

after the Canadian fires in August 2017 (Ansmann et al., ACP, 2018). The RFOV of the photometer is of the order of 0.5-1 degree (!!!). So, photometer observations suffer from strong multiple scattering effects. These effects are not corrected. However, many products are ok (e.g., size distribution, fits well to the lidar retrievals), several products are not ok. And SSA belongs to the questionable products. From 25-29 January 2020 the SSA (440nm, 670nm) was 0.96 to 0.99, and always >0.9 from 20-30 January. This is impossible as many simulations show. Even all the papers dealing with observed self lifting effects show that low SSA values are required to lift the smoke layers. So, we leave out to mention problematic AERONET SSA values.

L284: Please specify the wavelength

This is improved: 355nm and 532nm

L310: what do you mean by "coherent"? should it be 'persistent'?

We changed that:  to "persistent"

L356: Kloss2021b-- format of citation

We corrected that.

L363: "5-10 higher than"

Corrected.

L375: To be honest, the slight increase of the backscatter-related Angstrom exponent (532-1064) is not visible to me.  And if take into account the uncertainty of this parameter, I do not think you can derive a definitive increasing trend.

We do not fully agree and therefore write now: The less noisy Angstrom exponent values  for the long wavelength range (532-1064 nm) accumulate around 1.5 in January 2020 and are around 1.8 in the second half of 2020. This increase reflects a shift of the size distribution towards smaller particles. Larger particles may have been removed by sedimentation processes.

L393-396: Why not use the simultaneous Angstrom exponent measured by photometer?  You are converting columnar AOD to other wavelengths, so it should be converted with columnar Angstrom exponent, isn't it?

We checked that for some cases and found a difference of around 5%. We could thus change all this, but we leave it as is.

L415-419: As the smoke plume intensities decrease with time, the detection becomes more and more difficult, especially for plumes in the UTLS. In the second half year of 2021, mostly extinction coefficients of the plumes are under 2.5 Mm-1 and the signal in the cross channel got weaker and weaker because as you said the particle depolarization ratio decreases with time. Under this situation, are the low values of depolarization ratios (close to zero) reliable, how about the uncertainty? In Figure 8c, several data points show slightly negative depolarizations, such as 2020/01/02, 2020/06/25 and 2020-10-28, how come?

We checked all the critical data. The positive low depol values are still ok. We leave them in. However, in the case of the negative values, we finally had to accept: The data are simply too noisy. We removed them.

L514-522: In my opinion, the contrast of smoke depolarization ratios in Siberian fires and in ANYSO and PNE wildfires cannot be simply explained by the lifting time. The differences in particles size, absorption…, burning materials…could also contribute and should be mentioned in the text.

We still believe that the lifting process is essential and creates the difference! However, the reviewer is not wrong. So, we improved the discussion by adding some arguments:

This strong contrast to the ANYSO and PNE depolarization features again points to the fact that different lifting processes took place. PyroCb convection was responsible for fast lifting of smoke towards stratospheric heights (within less than a few hours) in the case of PNE and ANYSO so that emitted particles retained their irregular shape. On the other hand, slow ascent over days caused by self-lifting prevailed in the case of SILBE smoke. The aging process could be completed within these few days so that spherical shapes of the core-shell particles dominated. Additional effects can influence the optical properties, especially the lidar ratio, such as fire type, fuel material and meteorological conditions as mentioned above and in \citet{Ohneiser2020} and \citet{Ansmann2021}. A significant fraction of the slowly lifted smoke particles may be spherical tarballs forming from the emitted gases at low heights about 3-6 hours after injection \citep{China2013, Sedlacek2018, Adachi2019, Yuan2021}.

L574: citation format: remove the parenthesis (Stone et al., 2021)

Corrected.

L600: PSC not defined

Is now improved.

In Figure 4, we see the extinction-related Angstrom exponent is positive (i.e. ext 355 is higher than ext 532), while in Figure 5, it is the reverse, does it mean that particles are getting bigger or it is the uncertainty of measurement? Can you comment?

Sorry! This is our mistake. We already realized in January 2020 (Ohneiser et al., 2020) that the 387 nm Raman signals were only ok (for extinction retrieval) during the first fire smoke days (until 10-11 January 2020). But, in Fig 5 we obviously forgot this! The improved Fig 5 does no longer contain 355 nm extinction and lidar ratio values. More details, below.

Figure 1(b, c, d): The coastlines in the maps are difficult to read, please change the color and make them more visible. Explain the label 'CALIOP ANY' in Figure 1e.

This is now improved!

Figure 4: What are the criteria for the determination of reliable (solid line) and unreliable (dotted line) extinction coefficient? If the extinction coefficient above 14 km is too noisy to be reliable, so is the lidar ratio.

We want to avoid a lengthy discussion….. We found that the 355 nm extinction values above 14 km are not too bad (at least they are to some extent reasonable), so we show the less trustworthy 355

nm extinction values. However, we should avoid that in the case of the lidar ratio. So, now we only show the 'good' 355nm lidar ratio up to 14 km height.

Figure 5: The same to Figure 5: the criteria of determining trustworthy extinction should be given.

We improved Figure 5 by removing the 355 nm extinction and lidar ratio profiles. The rest is ok. The quality of the measurements is indicated by uncertainty bars.

Figure 6: Please specify the wavelength of the extinction coefficient (in Figure a and b) in the caption and in the text.

This is done.

Figure 8: The intensive parameters are all vertically varying, how was the average performed, daily and vertically averaged?   On 26 Jan 2020, the extinction-to-backscatter ratio was < 50 sr, but this data point is not seen in Figure 8b. In addition, the temporal resolution of data in Figure 8a is not the same with those in Figure 8b and 8c, why? Is it due to different quality control strategy?

We write now in the figure caption: Smoke layer mean optical properties (daily values) …

That means, vertically averaged values per day are shown. We checked the 26 Jan data point and included the value.

In Fig 8b and 8c, all analyzed DAYS are shown for which the analysis was successful (i.e., trustworthy, reasonable…), and also that appropriate AERONET observations were available. Proper days with good lidar and good AERONET observations were obviously rare. We do not state that explicitly.

Figure 13: The legend is missing

Legend in a) added. Legend in b not required: colors of x-axis replace the legend.

We finally went through the entire Section 4 (smoke impact on ozone depletion) to improve all explanations and to be more precise regarding notation. We added a new reference of a recently completed manuscript, submitted a few days ago:

Ansmann et al., 2022:

Ansmann, A., Ohneiser, K., Chudnovsky, A., Knopf, D. A., Eloranta, E. E., V. D., Seifert, P., Radenz, M., Barja, B., Zamorano, F., Jimenez, C., Engelmann, R., Baars, H., Griesche, H., Hofer, J., Althausen, D., and Wandinger, U.: Ozone depletion in the Arctic and Antarctic stratosphere induced by wildfire smoke, Atmos. Chem. Phys. Disc., 22, https://doi.org/10.5194/acp-2022-XXX, 2022.